# Early progression to active tuberculosis is a highly heritable trait driven by 3q23 in Peruvians

Yang Luo [1,2,3,4,5], Sara Suliman [1], Samira Asgari [1,2,3,4,5], Tiffany Amariuta [1,2,3,4,5,6], Yuriy Baglaenko[1,2,3,4,5], Marta Martínez-Bonet [1], Kazuyoshi Ishigaki[1,2,3,4,5], Maria Gutierrez-Arcelus [1,2,3,4,5], Roger Calderon[7], Leonid Lecca[7], Segundo R. León[7], Judith Jimenez[7], Rosa Yataco[7], Carmen Contreras[7], Jerome T. Galea[8], Mercedes Becerra[9], Sergey Nejentsev[10,11], Peter A. Nigrovic[1,12], D. Branch Moody [1], Megan B. Murray[9] & Soumya Raychaudhuri [1,2,3,4,5,13]

Of the 1.8 billion people worldwide infected with *Mycobacterium tuberculosis*, 5–15% will develop active tuberculosis (TB). Approximately half will progress to active TB within the first 18 months after infection, presumably because they fail to mount an effective initial immune response. Here, in a genome-wide genetic study of early TB progression, we genotype 4002 active TB cases and their household contacts in Peru. We quantify genetic heritability ($h_g^2$) of early TB progression to be 21.2% (standard error 0.08). This suggests TB progression has a strong genetic basis, and is comparable to traits with well-established genetic bases. We identify a novel association between early TB progression and variants located in a putative enhancer region on chromosome 3q23 (rs73226617, OR = 1.18; $P = 3.93 \times 10^{-8}$). With in silico and in vitro analyses we identify rs73226617 or rs148722713 as the likely functional variant and *ATP1B3* as a potential causal target gene with monocyte specific function.

[1] Division of Rheumatology, Inflammation and Immunity, Brigham and Women's Hospital, Harvard Medical School, Boston, MA, USA. [2] Division of Genetics, Brigham and Women's Hospital, Harvard Medical School, Boston, MA, USA. [3] Broad Institute of MIT and Harvard, Cambridge, MA, USA. [4] Department of Biomedical Informatics, Harvard Medical School, Boston, MA, USA. [5] Center for Data Sciences, Brigham and Women's Hospital, Harvard Medical School, Boston, MA, USA. [6] Graduate School of Arts and Sciences, Harvard University, Cambridge, MA 02138, USA. [7] Socios En Salud, Lima, Peru. [8] School of Social Work, University of South Florida, Tampa, FL, USA. [9] Department of Global Health and Social Medicine, and Division of Global Health Equity, Brigham and Women's Hospital, Harvard Medical School, Boston, MA, USA. [10] Department of Medicine, University of Cambridge, Cambridge, UK. [11] Department of Molecular Cell Biology and Immunology, Amsterdam University Medical Centers, Amsterdam, Netherlands. [12] Division of Immunology, Boston Children's Hospital, Boston, MA, USA. [13] Arthritis Research UK Centre for Genetics and Genomics, Manchester Academic Health Science Centre, University of Manchester, Manchester, UK. Correspondence and requests for materials should be addressed to M.B.M. (email: megan.murray.epi@gmail.com) or to S.R. (email: soumya@broadinstitute.org)

The infectious pathogen *Mycobacterium tuberculosis (M.tb)* infects about one-quarter of the world's population[1]. Approximately 5–15% of infected individuals progress to active TB while the vast majority remain infected with viable latent *M.tb* (Fig. 1a). In 2017, approximately 10 million new patients were diagnosed with active TB, and 1.6 million people died from TB-related diseases[2]. Active TB can develop immediately (within the first 18 months) after recent *M.tb* infection or after many years of latency, presumably caused via distinct disease mechanisms. Late progression or TB reactivation is more likely the consequence of acquired immune compromise due to other diseases or ageing, whereas early progression is presumably due to failure in mounting the initial immune response that contains the bacterial spread[3]. Previous studies have indicated a strong heritable component of population-wide TB susceptibility, that includes early disease progression, reactivation, and infection[4–6]. But whether early progression has a different genetic architecture compared to population-wide susceptibility has yet to be defined.

Reported associations for TB and other infectious diseases have to be considered in the context of TB diagnostic criteria and selected control groups[7,8]. To date genome-wide association studies (GWAS) of TB have compared mixed pools of TB patients with early progression or reactivation, to population controls, who may not have been exposed to *M.tb* at all[9–13]. Hence, known human genetic loci associations with clinical outcomes might represent risk factors for *M.tb* infection, progression from recent *M.tb* exposure to active TB, or reactivation of TB after a period of latency. Infection, progression, and reactivation represent pathophysiologically distinct disease transitions likely involving distinct mechanisms of transmission, early innate immune response, and control by adaptive immunity. Thus, the study of mixed TB populations using controls of unknown exposure status may underestimate or miss genetic associations for these separate stages of disease.

In this study, we perform a genome-wide association study of a large sample of early TB progression cases (2175 recently exposed cases and 1827 controls). We first establish early TB progression has a strong genetic basis that is comparable to other complex traits. We further identify a novel association with early TB progression, prioritize likely causal variants and functional genes, and propose new candidate mechanisms of host response in early TB progression.

## Results
**Building an early progression to active tuberculosis cohort**. To identify host factors that drive pulmonary early TB progression, we conducted a large, longitudinal genetic study in Lima, Peru (Fig. 1b), where the TB incidence rate is one of the highest in the region[14]. We enrolled patients with microbiologically confirmed pulmonary TB. Within 2 weeks of enrolling an index patient, we identified their household contacts (HHCs) and screened for infection as measured by a tuberculin skin test (TST) and for signs and symptoms of pulmonary and extra-pulmonary TB. HHCs were re-evaluated at 2, 6, and 12 months. We considered individuals to be early progressors if they are (1) index patients whose *M.tb* isolates shared a molecular fingerprint with isolates from other enrolled patients; (2) HHCs who developed TB disease within 1 year after exposure to an index patient and (3) index patients who were 40-years old or younger at time of diagnosis. We considered HHCs who were TST positive at baseline or any time during the 12 month follow up period, but who had no previous history of TB disease and remained disease free, as non-progressing controls (Methods, Fig. 1b). In total, we genotyped 2175 recently exposed pulmonary TB cases (early progressors) versus 1827 HHCs with latent tuberculosis infection, who had not progressed to active TB during 1 year of follow-up (non-progressors), as controls (Methods, Supplementary Table 1).

**Genomic analysis demonstrates the distinct genetic heritage of Peruvians**. Peru is a country with a complex demographic history and underexplored genomic variation. When Spanish conquistadors arrived in the region in the 16th century, Peru was the center of the vast Inca Empire and was inhabited by a large Native American population[15,16]. During the colonial period, Europeans and Africans (brought in as slaves) arrived in large numbers to Peru. After Peru gained its independence in 1821, there was a flow of immigrants from southern China to all regions of Peru as a replacement for slaves[17,18]. As a result, based on the analysis of our large genomic cohort, the genetic background of the current Peruvian population is shaped by different levels of admixture between Native Americans, Europeans, African and Asian immigrants that arrived in waves with specific and dated historical antecedents. When compared to individuals from other South American countries[19,20], Peruvians tend to share a greater genetic similarity with Andean indigenous people such as Quechua and Aymara (Fig. 2, Supplementary Fig. 1, Methods).

This unique genetic heritage provides both a challenge and an opportunity for biomedical research. To optimally capture genetic variation, and particularly rare variations in Peruvians, we designed a 712,000-SNP customized array (LIMAArray) with genome-wide coverage based on whole-exome sequencing data from 116 active TB cases (Methods, Supplementary Table 2, Supplementary Fig. 2). When compared to other more comprehensive genotyping platforms available at the time, LIMAArray showed an ~5% increase in imputation accuracy, particularly for population-specific and low-frequency variants (Supplementary Table 3). We derived estimated genotypes for ~8 million variants using the 1000 Genomes Project Phase 3[19] as the reference panel and tested single marker and rare-variant burden associations with linear mixed models that account for both population stratification and relatedness in the cohort (Supplementary Figs. 3–4, Methods). Genome-wide association results of 2160 cases and 1820 controls after quality control (Methods) are summarized in Supplementary Fig. 5. We observed no inflation of test statistics ($\lambda_{GC} = 1.03$, $\lambda_{GC} = 1.00$ for common and rare association analyses respectively), which suggests potential biases were strictly controlled in our study. We observed no significant rare variant (minor allele frequency (MAF) <1%) association with TB progression after performing gene-based generalized linear mixed model (Methods).

**Progression of recent *Mycobacterium tuberculosis* exposure to active tuberculosis is a highly heritable complex trait**. To investigate the genetic basis of early TB progression, we first estimated its variant-based heritability ($h_g^2$). Using GCTA[21] we estimated $h_g^2$ of TB progression to be 21.2% (standard error (s.e.) = 0.08, $P_{Gaussian} = 2.64 \times 10^{-3}$) on the liability scale with assumed incidence rate of 0.05 in the cohort (Methods). To avoid biases introduced from calculating genetic relatedness matrices (GRMs) in admixed individuals, we calculated two different GRMs based on admixture-aware relatedness estimation methods[22,23] and removed related individuals. Both admixture-aware methods reported similar $h_g^2$ estimates (Supplementary Table 4), indicating our reported heritability estimation is robust under different model assumptions. We quantified $h_g^2$ of TB progression and observed a surprisingly strong genetic basis. This degree of heritability is comparable to traits with a well-established genetic basis (Supplementary Table 5). For example,

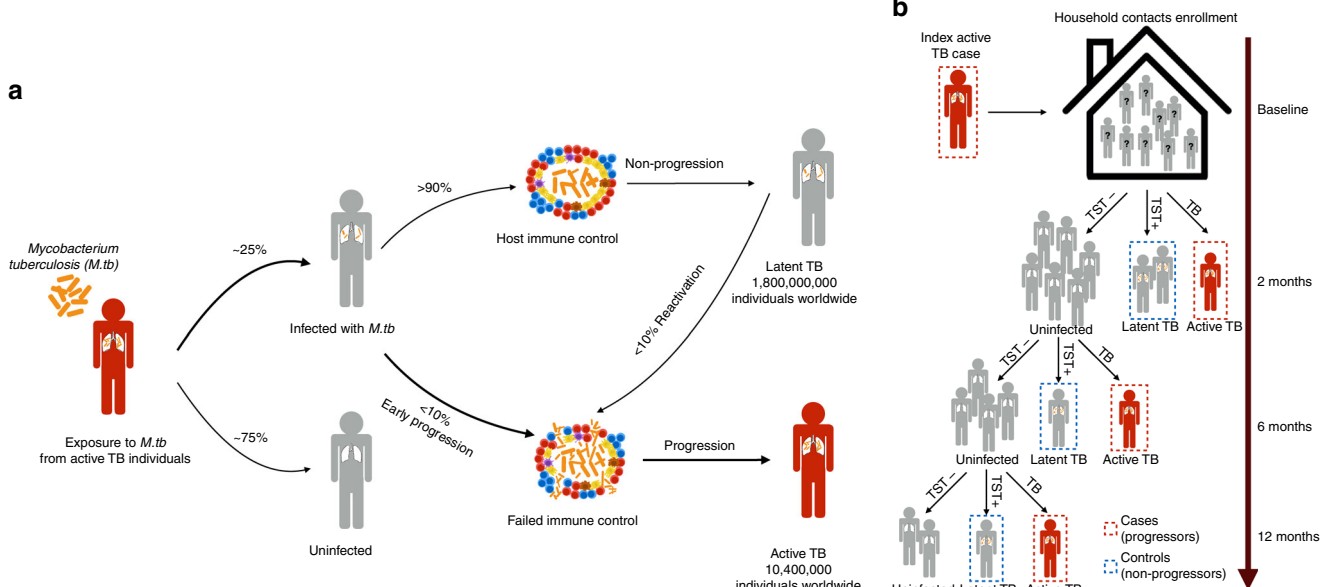

**Fig. 1** Overview of phases of *Mycobacterium tuberculosis* (*M.tb*) infection and sample collection. **a** Pathophysiology of TB. Major steps following from the initial exposure to *M.tb* are outlined, with the percentages of individuals progressing between steps taken from the WHO TB report[14]. **b** Schema of cohort collection. In this study, we focus on a genetic study between recently exposed active pulmonary TB cases (progressors) and subjects with tuberculin skin test (TST) positive results, who did not progress to active TB (non-progressors). Index cases had sputum with confirmed TB. Controls were recruited in the same household as index cases, with 12 month follow up periods to confirm infection status using TST

GWAS have identified ~200 risk loci for Crohn's disease[24,25], which has a reported $h_g^2$ of 28.4% (s.e. = 0.02, $P_{Gaussian} = 8.62 \times 10^{-71}$)[24]. To compare the genetic heritability between early TB progression and population-wide TB susceptibility, we subsequently obtained genotypes from a previous TB study conducted in Russia with 11,137 individuals[11]. Using GCTA, we estimated the $h_g^2$ of population-wide TB susceptibility to be 17.8% (s.e. = 0.02, $P_{Gaussian} = 2.85 \times 10^{-21}$) with assumed prevalence of 0.04[26]. Even though the point estimate of $h_g^2$ of TB progression is greater than that of population-wide TB risk in the Russian study, these estimates are not statistically different from each other (two-tailed *t*-test $P = 0.68$, Supplementary Fig. 6). Regardless, the strong host genetic basis of TB progression suggests that larger progression studies may be well-powered to discover additional variants.

**Genome-wide association study identifies a novel association at 3q23**. We next identified a novel risk locus associated with TB progression on chromosome 3q23, which is comprised of 11 variants in non-coding regions downstream of *RASA2* and upstream of *RNF7* ($P < 1 \times 10^{-5}$) (Fig. 3a, Supplementary Table 6, Dataset 1). The strongest association with early TB progression was at a genotyped variant rs73226617 (OR = 1.18; $P = 3.93 \times 10^{-8}$). To test for artifacts and to identify stronger associations that might have been missed due to genotyping and imputation, we first checked the genotype intensity cluster plot of the top associated variant which showed a clear separation between genotypes AA, AG, and GG (Supplementary Fig. 7). We then designed individual TaqMan genotyping assays for four top associated variants (Methods, Supplementary Table 7). We genotyped these four SNPs in 4002 initial subjects and concluded that all four variants show a high concordance rate (>99%) with imputed genotypes (Supplementary Table 6, Dataset 1). Because all 11 variants in the risk locus are in high linkage disequilibrium (LD) with each other (Supplementary Fig. 8), the other imputed variants are also likely to have high imputation quality.

To determine whether the reported risk locus at 3q23 also has an independent association with TB progression from recent *M.tb* infection, we conducted a case-only analysis removing age from our case selection criteria. This approach is based on the premise that TB cases that share a DNA fingerprint for *M.tb* and HHCs who developed active TB are epidemiologically related while cases in which *M.tb* fingerprints are different might have resulted from remote infection that reactivated during the study assessment[27]. 1472 out of 2175 presumed early progressors shared molecular fingerprint of *M.tb isolates* with another case or developed active TB during the 1 year of follow-up (Supplementary Fig. 9). Other cases did not have a shared molecular fingerprint among *M.tb* isolates or did not come from the same household as the index case, leading to a lower degree of certainty in the early progression status of these cases. In this case-only analysis, the top associated signal rs73226617 was nominally associated with early progression (OR = 1.09, $P = 0.016$). A heritability analysis restricted to those that shared the same molecular fingerprint or from the same household estimated in a $h_g^2$ of 22.1% (s.e. = 0.06, $P_{Gaussian} = 1.32 \times 10^{-4}$) despite the smaller number of samples. To assess the independence of the stratified cases compared to the overall case-control analysis, we first compared reported effect sizes in both analyses and observed a low Pearson correlation ($r = 0.014$, Supplementary Fig. 10). To test the significance of the reported association, we performed a permutation analysis, where we randomly permuted the case/control status in the stratified analysis. After permuting for 10,000 times, the observed OR (1.09) has a $P$-value of 0.017 (Supplementary Fig. 11). We next performed a Bayesian analysis to test whether the reported association is restricted to the early progressors after recent exposure to *M.tb*. (Methods). The disease specific approximate Bayes Factor[28] (i.e., the ratio of the marginal likelihood for a model where the variant is only associated with early progressor who has a shared molecular fingerprints and/or a secondary cases and for a model where is associated with all progressors) is 0.42. This suggested that the SNP is most likely to be associated with early progressors who have recent exposure to *M.tb*. alone, but

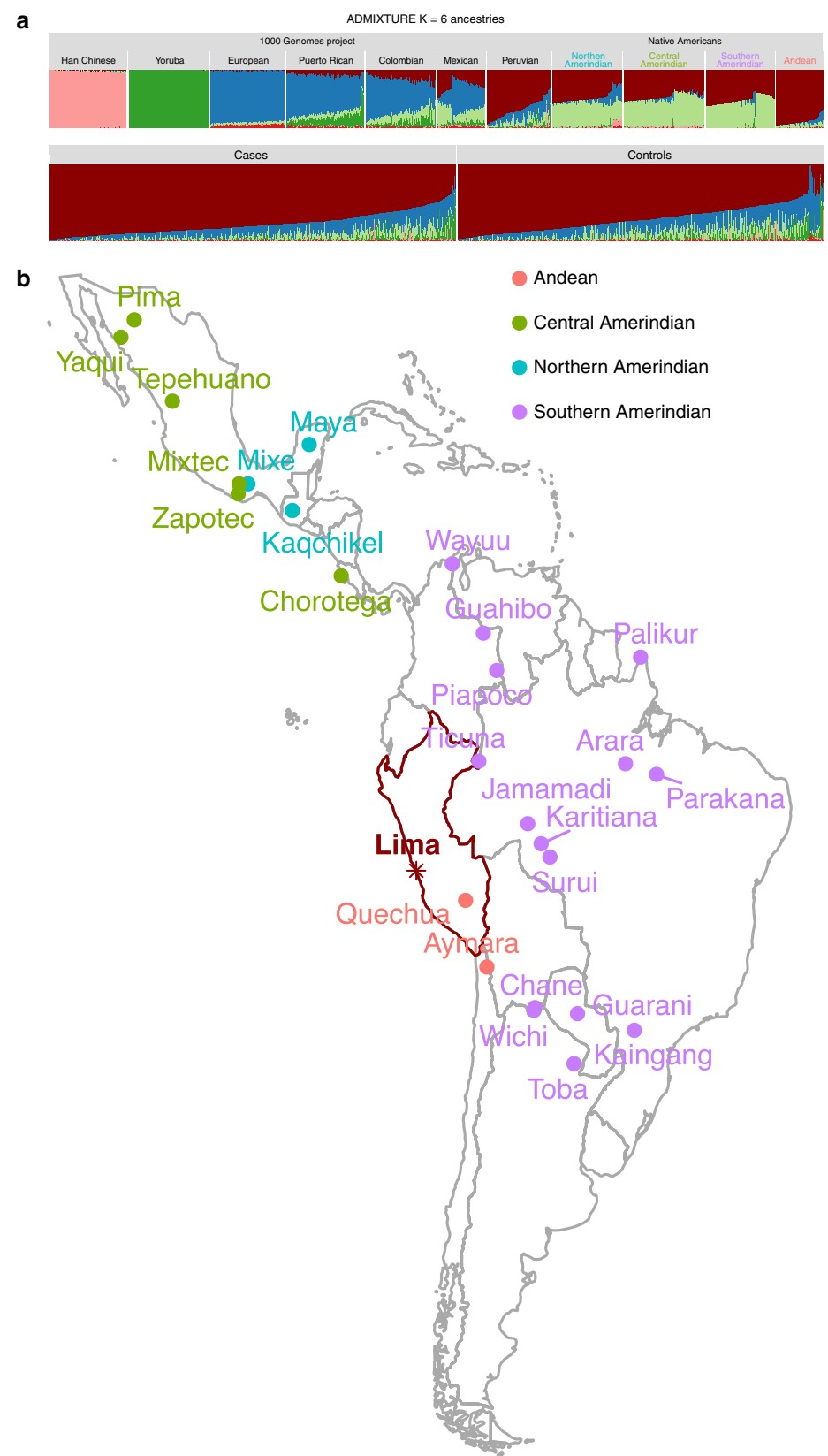

almost equally likely to be associated with TB progression in general.

We examined the 11 most associated variants for early TB progression identified in the Peruvian cohort in previously published GWAS datasets[9–11,29] (Supplementary Table 8, Dataset 2). These SNPs were less frequent (<1%) in the African populations than in the European and Peruvian populations, resulting in lower statistical power to detect association. We

**Fig. 2** Global ancestry analysis of Peruvian populations. **a** ADMIXTURE plot of admixed individuals and continental reference panels. Each individual is represented as a thin vertical bar. The colors can be interpreted as different ancestries. Reference panels are either from the 1000 Genomes project[19] (1000G) or Native American individuals collected from Reich et al. *Nature*[20]. Han Chinese are from Beijing, China; Yoruba are from Ibadan, Nigeria; European individuals are Utah Residents (CEPH) with Northern and Western European Ancestry; Puerto Ricans are from Puerto Rico; Colombians are from Medellin, Colombia; Mexican individuals are from Los Angeles, California; Peruvians are from Lima, Peru. Northern Amerindian includes individuals from Maya, Mixe, and Kaqchikel. Central Amerindian includes individuals from Pima, Zapotec, Mixtec, Yaqui, Chorotega, Tepehuano. Southern Amerindian includes individuals from Piapoco, Karitiana, Surui, Wayuu, Jamamadi, Parakana, Guarani, Kaingang, Ticuna, Palikur, Toba, Arara, Wichi, Chane and Guahibo. Andean population includes Quechua and Aymara. $K = 6$ models are shown above, $K = 3$ through $K = 15$ models are available in Supplementary Fig. 1. Source data are provided as a Source Data file. **b** Map of locations of sampled Native American populations[20]

therefore examined the SNPs in two previously published Russian[11] (5530 TB cases and 5607 controls) and Icelandic[29] (4049 TB cases and 6543 TST + controls) GWAS datasets. We observed that the effects in the Russian cohort were similar, as they shared comparable ORs of 1.10 (Russian) and 1.18 (Peru) for rs73226617 ($P_{Russia} = 0.065$). In contrast, there was no signal observed in the Icelandic cohort (OR = 1.06, $P_{Iceland} = 0.437$). Consistent with our previous case-only analysis, the weaker signals observed in both European cohorts indicate that 3q23 is associated with early TB progression. The lack of association observed in the two European cohorts could be due to the inclusion of reactivation TB cases and noninfected controls; differences in TB prevalence (Methods).

We next examined how previously published TB GWAS risk loci are associated with progression in this study. We detected evidence of association in a previously reported TB locus at rs9272785 in the HLA region[29] (OR = 1.04, $P = 4.49 \times 10^{-3}$), but did not detect signals at other reported risk loci (Supplementary Table 9). Thus, previously reported loci may relate to infection or reactivation phenotypes, rather than early TB progression whereas HLA association may affect both early progression and reactivation. Next, we performed an HLA imputation using a multi-ethnic HLA reference panel (Methods), and obtained genotypes for classical alleles as well as amino acid positions of three class I (*HLA-A, HLA-B, HLA-C*) and three class II (*HLA-DQA1, HLA-DQB1, HLA-DRB1*) HLA genes. Using the same linear mixed model framework (Methods, Supplementary Fig. 12), we tested associations between specific amino acid positions and TB progression which identified the most significant association at amino acid position 73 of *HLA-A* (OR = 1.12, $P = 1.03 \times 10^{-6}$). We noted several other amino acids of class I genes with suggestive associations ($P < 1 \times 10^{-5}$) including position 97 of *HLA-B* (OR = 1.05, $P = 8.99 \times 10^{-6}$). Notably, amino acid variability at this position affects the structure and flexibility of the peptide-binding groove and is associated with many infectious and autoimmune phenotypes, such as HIV-1 viral load[30,31] and ankylosing spondylitis[32]. These results suggest that HLA class I genes might play a role in TB progression.

To try to identify which of the variants in our reported risk locus is likely to be the functional polymorphism affecting the risk of pulmonary TB progression, we employed the FINEMAP[33] software (Methods). The 90% credible set includes seven genomic variants, with rs73226617 having the highest posterior probability (0.54), followed by rs58538713 (0.16) and the indel rs148722713 (0.05) among 713 variants in the region (Fig. 3b, Supplementary Table 6, Dataset 1).

**A monocyte-specific regulatory element in 3q23 is implicated in TB progression**. To identify likely functional variants and target genes, we employed a method called IMPACT (Inference and Modeling of phenotype-related ACtive Transcription)[34]. Briefly, IMPACT identifies regions predicted to be involved in transcriptional regulatory processes related to a key transcription factor of a cell type (Methods) by leveraging information from

approximately 400 chromatin and sequence annotations in public databases (Fig. 3c, Supplementary Table 10, Dataset 3). Each variant is assigned a probability between 0 (least likely to be a regulatory element) and 1 (most likely to be a regulatory element). We tiled through the 23,308 base pair region on a per-nucleotide basis, computing the probability of a cell-type regulatory element separately for 15 different cell types and cell states of which 10 are immune cell types with known roles in TB outcomes, including T cells, B cells, monocytes, macrophages, and peripheral blood cells (Fig. 3e). We observed monocyte-specific predicted regulatory elements at rs73226617 and rs148722713 (IMPACT score 0.79 and 0.41, respectively, Fig. 3d).

We next searched for other epigenomic evidence that may indicate changes in transcriptional enhancers and other cis-regulatory elements. Given the possible monocyte-specific activity of the identified risk locus, we actively sought datasets that include monocyte primary cells or cell lines. We used data presented in the BLUEPRINT project[35] to search for chromQTLs. We observed significant chromQTL present in the region (characterized by the presence of H3K4me1) in monocytes (Supplementary Fig. 13) suggesting that this region indeed has an active enhancer. The rs73226617 SNP was included in this region, but did not itself have evidence of chromQTL activity; however, it is in high LD with the top associated chromQTL signal (rs1568171, D' = 1.0).

Based on the IMPACT analysis and the suggested enhancer activity in monocytes, we studied monocytic cells (THP1) as the most likely experimental model for locus-specific gene regulatory activity. We performed electrophoretic mobility shift assays (EMSA) to test whether the variants differentially bound nuclear complexes in an allele-specific manner among the seven variants that constitute the 90% credible set (Methods). We could detect differential protein binding that was competed out by unlabeled probes for three of the risk alleles (rs73226617, rs58538713, and rs148722713) (Supplementary Fig. 14), providing evidence that these alleles might confer differential transcription factor binding activity, and in the right context may lead to altered enhancer activity.

On the basis of posterior probabilities from the genetic data, EMSA binding assays demonstrating the capacity to alter binding of nuclear extract protein, and localization to an enhancer region with regulatory potential, we identified rs73226617, and rs148722713 as the most likely causal alleles.

**Potential target genes implicated by the TB progression risk locus**. Next, we searched public promoter Hi-C databases[36,37] to identify any significant interactions between the monocyte-specific enhancer harboring our most likely causal allele, rs73226617 and rs148722713. We found that in monocytes, both of the risk variants (rs73226617, rs148722713) are in a region that interacts with the promoter of *ATP1B3* (Supplementary Fig. 15a, b). Similar to the IMPACT results, we found the variant-gene interactions are strongest in monocytes compared to other cell types (Supplementary Fig. 15c, d), suggesting cell-type-specific activities in the

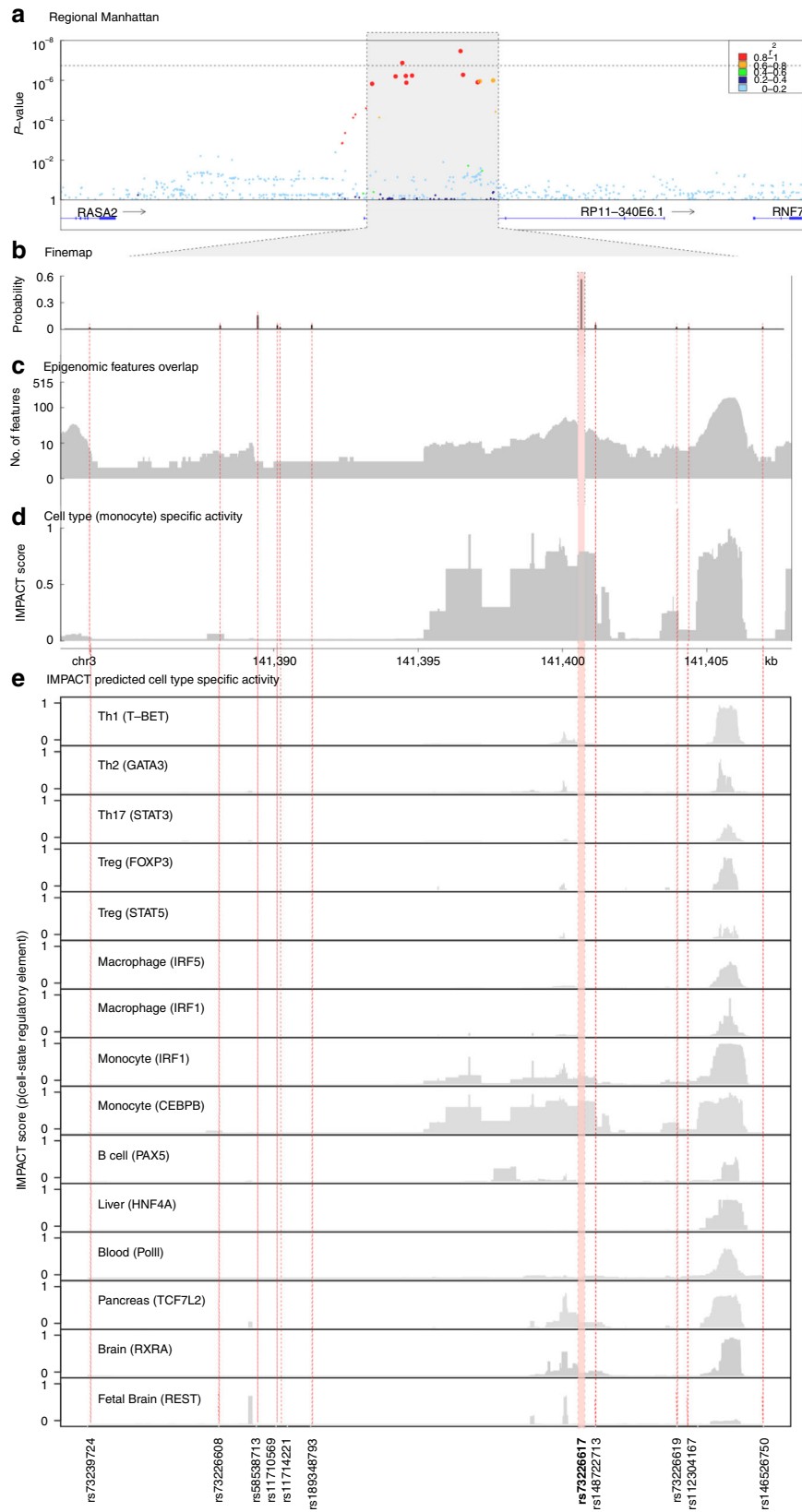

identified TB risk locus. *ATP1B3* (ATPase Na+/K+ Transporting Subunit Beta 3) is a protein-coding gene, which belongs to the family of Na+/K+ and H+/K+ ATPases. Na+/K+–ATPases are composed of an alpha, beta, and FXYD subunits, are integral membrane proteins responsible for establishing and maintaining the electrochemical gradients of sodium and potassium ions across the plasma membrane through active transport against their osmotic gradients. A recent study demonstrated that the Na,

**Fig. 3** Genome-wide association details of the 3q23 locus. **a** A regional association plot of the 3q23 locus including all genotyped and imputed variants. The horizontal line indicates the genome-wide significant threshold at $1.78 \times 10^{-7}$ for Peruvian populations (Methods). **b** Fine-mapping posterior probability of all variants in the chr3:140221602-145217859 region. **c** Number of overlaps between all variants in the risk locus and ~400 epigenetic features. **d** Predicted posterior probability of cell-type-specific gene regulatory activity using IMPACT (Inference and Modeling of phenotype-related ACtive Transcription) based on the epigenetic chromatin signature of binding sites of the transcription factor CEBPB in monocytes. **e** Intersection of nucleotide-resolution of variant cell-state IMPACT annotations with potential causal variants in 3q23 locus. The y-axis shows the posterior probability of predicted cell-state regulatory activity among each variant in 15 different cell types and cell states. The x-axis shows the genomic positions of all variants among the identified risk locus. The bolded variant (rs73226617) is the leading risk variant from the association study which shows the highest predicted cell-state regulatory activity in monocytes (masked by CEBPB transcription factor). Dashed lines highlight 11 top associated variants. Genotyped variant rs73226617 is highlighted in red. Source data are provided as a Source Data file

K ATPase Beta 3 subunit in monocytes has an important function in mediating a normal T cell response[38]. Indeed ligating it with an antibody resulted in a blunted T cell response after stimulation. This effect was specific to the monocytes population. Consistent with these findings, differential expression of *ATP1B3* in whole blood, along with genes coding for other members of the Na+/K +-ATPases, was recently reported to be associated with TB progression in an African cohort of household contacts of TB patients[39]. Collectively, the Hi-C analysis and reported association with TB progression point to *ATP1B3* as a candidate gene of the risk locus in 3q23.

Since in silico evidence suggested that our identified TB risk locus harbors monocyte-specific regulatory elements, we used the CRISPR/Cas9 system to introduce insertions/deletions around the top associated variant rs73226617 (Methods, Supplementary Fig. 16a). Among 23 sorted and grown clones that had unchanged risk loci or harbored unique edits and deletions (Supplementary Table 11 and Supplementary Fig. 16b, c), we did not observe differential gene expression between edited and unedited THP1 clones in the *cis*-genes around the 500 kb window of the leading rs73226617 variant (ANOVA *P*-value > 0.05, Methods, Supplementary Fig. 17a). This CRISPR/Cas9 approach to disrupt the putative enhancer has multiple limitations. Firstly, while we observed no effect in THP1 cell lines, this might result from differences between primary monocytes and transformed THP1 cell lines, or failure to identify the relevant activation conditions and cell context to test enhancer activities, which are known to influence eQTL interactions[40–43]. Secondly, although we chose guide RNA sequences optimized to target the 3q23 region, and did not identify other likely genomic targets by nucleotide homology, off-target effects are still possible (Supplementary Table 20). We analyzed independent edited THP1 clones to reduce the likelihood that we propagated additional off-target genomic edits. However, a genome-wide analysis of differential expression also did not detect any other differentially expressed targets outside the local neighborhood surrounding rs73226617 (Supplementary Fig. 17b), suggesting that off-target disruptions were unlikely. In particular, we noted the enhancer activity seen in primary monocytes, is not seen in THP1 cell lines[44–47] (Supplementary Fig. 18).

## Discussion
Overall, our results argue that rapid TB progression is a highly heritable trait, comparable to other human diseases with an established genetic origin. More generally, these results begin to address general questions about genomic approaches to infectious diseases, which have lagged behind other diseases and complex traits in terms of locus discovery (Supplementary Table 5). Infections, especially chronic infectious diseases, play out in highly distinct phases that involve exposure, crossing epithelial boundaries, pathogen expansion, locating a host niche, and in the case of TB, decades-long persistence, reactivation, and re-transmission. Each of these stages can be controlled by distinct

host factors. Our analysis indicates that progression from recent *M.tb* exposure to active TB has a different genetic basis compared to TB reactivation. Specific analysis of clinical progression as a distinct phase allows for a more powerful detection of risk factors for an equal number of samples, as compared to case-control studies, which are an amalgamation of different phenotypes. Thus, this work suggests that while detailed, stage-specific phenotypic profiling may be more costly, it may offer key advantages for infectious disease genetic studies. Specifically, it allows for precise phenotype definitions and identification of biological targets with specific implications. Therefore, detailed phenotypic profiling should become an additional valuable approach for future genetic studies of infectious diseases. Detailed phenotyping enables investigators to dissect pathogenic mechanisms at different stages of infection and disease progression.

## Methods
**Ethics statement.** We recruited 4002 subjects from a large catchment area of Lima, Peru that included 20 urban districts and ~3.3 million residents to donate a blood sample for use in our study.

We obtained written informed consent from all the participants. The study protocol was approved by the Institutional Review Board of Harvard School of Public Health and by the Research Ethics Committee of the National Institute of Health of Peru.

**Preparation of genome-wide genetic data.** We enrolled index cases as adults (aged 15 and older) who presented with clinically suspected pulmonary TB at any of 106 participating health centers. We excluded patients who resided outside the catchment area, who had received treatment for TB before and those who were unable to give informed consent. Pulmonary TB patients have been diagnosed by the presence of acid fast bacilli in sputum smear or a positive *M.tb* culture at any time from enrollment to the end of treatment. All cultures of the index cases were genotyped using mycobacterial interspersed repetitive units-variable number of tandem repeats (MIRU-VNTR). Within 2 weeks of enrolling an index patient, we enrolled his or her household contacts (HHCs). The *M.tb* status was determined using the Tuberculin Skin Test (TST). All HHCs were evaluated for signs and symptoms of pulmonary and extra-pulmonary TB disease at 2, 6, and 12 months after enrollment. All cases were HIV-negative, culture-positive and drug-sensitive who have pulmonary TB. We defined cases who were likely to have recently exposed TB, if a case satisfied at least one of the three criteria: (1) exposed HHCs who developed active TB during a 12 month follow up period; (2) index patients whose *M.tb* isolates shared a molecular fingerprint with isolates from other enrolled patients and (3) index patients who were 40-years old or younger at time of diagnosis. To maximize the likelihood that controls were exposed to *M.tb* but did not develop active disease, we chose them from among TST positive HHCs with no previous history of TB disease, and who remained disease free at the time of recruitment both by directly re-contacting individuals to inquire about their latest medical history and by checking their names against lists of notified TB patients at all of the 106 health clinics. Where possible, we chose controls who are less than second-degree related to the index cases.

**Customized axiom array for Peruvian populations.** We developed a custom array (LIMAArray) based on whole-exome sequencing data from 116 active TB cases to optimize the capture of genome-wide genetic variation in Peruvians. Many markers were included because of known associations with, or possible roles in, phenotypic variation, particularly TB-related (Supplementary Table 12). The array also includes coding variants across a range of minor allele frequencies (MAFs), including rare markers (<1% MAF), and markers that provide good genome-wide coverage for imputation in Peruvian populations in the common (>5%), low frequency (1–5%) and rare (0.5–1%) MAF ranges (Supplementary Table 3). This

approach allowed the detection of rare population-specific coding variants and those which predisposed individuals to TB risk.

**Genotyping and quality control**. We extracted genomic DNA from whole blood of the participating subjects. Genotyping of all samples was performed using our customized Affymetrix LIMAArray. Genotypes were called in a total of 4002 samples using the apt-genotype-axiom[48]. Individuals were excluded if they were missing more than 5% of the genotype data, had an excess of heterozygous genotypes (±3.5 standard deviations, Supplementary Table 13), duplicated with identity-by-state >0.9 or index cases with age at diagnosis greater than 40-years old. After excluding these individuals, we excluded variants with a call rate less than 95%, with duplicated position markers, those with a batch effect ($P < 1 \times 10^{-5}$), Hardy–Weinberg (HWE) $P$-value below $10^{-5}$ in controls, and a missing rate per SNP difference in cases and controls greater than $10^{-5}$ (Supplementary Table 14). In total, there were 3980 samples and 677,232 SNPs left for imputation and association analyses.

**Imputation and association analyses**. The genotyped data were pre-phased using SHAPEIT2[49]. IMPUTE2[50] was then used to impute genotypes at untyped genetic variants using the 1000 Genomes Project Phase 3 dataset[19] as a reference panel. For chromosome X, males are coded as diploid. That is male genotypes are coded as 0/2 and females genotypes are coded as 0/1/2. HLA imputation was performed using SNP2HLA[51] and a multi-ethnic HLA imputation reference panel[52]. Imputed SNPs were excluded if the imputation quality score $r^2$ was less than 0.4, HWE P-value < $10^{-5}$ in controls or a missing rate per SNP greater than 5%. After filtering, 7,756,401 SNPs were left for further association analyses.

Common single variant associations were tested with a linear mixed model (LMM) implemented in GEMMA[53] version 0.94.1 on genotype likelihood from imputation assuming an additive genetic model. We used the genetic relatedness matrix (GRM) as random effects to correct for cryptic relatedness and population stratification between collected individuals. Sex and age were included as fixed effects. The GRM was obtained from an LD-pruned ($r^2 < 0.2$), with MAF ≥ 1% after removing large high-LD regions[54] (Supplementary Table 15) dataset of 154,660 SNPs using GEMMA[55] version 0.7–1. To determine an appropriate genome-wide significant threshold for Peruvian populations, we followed the permutation strategy proposed by Kanai et al.[56], and considered a variant is significantly associated with TB progression, if it has a $P$-value smaller than $1.78 \times 10^{-7}$.

Gene-based rare variant (MAF < 1%) burden test was performed using GMMAT[55] version 0.7–1, a generalized linear mixed model framework. For each gene $j$, we aggregated the information for multiple rare variants into a single burden score ($C_i = \sum_{j=1}^{M} G_{ij}$) for each subject $i$. Where $G_{ij}$ denotes the allele counts {0,1,2} for m variants in the gene. The genomic control inflation factor ($\lambda_{GC}$) for variants after imputation was 1.03 and 1.00 for common and rare association study respectively (Supplementary Fig. 5), indicating that we have successfully controlled for any residual population structure or cryptic relatedness between genotyped samples.

To minimize false-positive signals due to population stratification and heterogeneity of effects due to differential LD in admixed populations, we also computed GRMs based on methods[22,23] that account for inflation of identity-by-state statistics due to admixture LD. LMM with admixture-aware GRMs resulted in numerically similar association statistics to those from unadjusted analyses (Supplementary Table 16). To control for the potential effect of ancestry differences between cases and controls and the robustness of our reported findings, we tested our linear mixed model adding Native American ancestry inferred from ADMIXTURE analysis (K = 6) as a covariate. We observed similar association strengths genome-wide (Supplementary Fig. 19) and in our reported top associations (Supplementary Table 6). This result supports that our reported associations are independent of individual ancestral proportions.

To identify likely causal variants in the identified risk locus, we used the FINEMAP[33] method to calculate marginal likelihoods and Bayes factor for each variant assuming that there is one true causal variant in the region, and it has been included in the analysis and has been well imputed (–n-causal-max 1). We used the in-sample LD scores calculated using LDstore[57] to further increase the accuracy of the fine-mapping analysis.

**TaqMan SNPs and genotyping**. Selection of SNPs in the 3q23 locus was conducted based on information from the dbSNP database (http://www.ncbi.nlm.nih.gov/projects/SNP/). Four polymorphisms rs73226617, rs73226619, rs73239724, and rs73226608 were included for the genotyping tests. Real-time PCR using the following calculations: 2.5 uL Genotyping Master Mix, 0.25 uL SNP Assay-probes, and 2.25 uL DNA template (at 5 ng/uL = 11.25 ng total).Thermal cycling conditions were as follows: 60 °C 30 s Pre-read, 95 °C for 10 min, followed by 40 cycles at 95 °C for 15 s and at 60 °C for 1 min, then 60 °C 30 s Post-read. Genotyping of the polymorphisms was carried out using the 5' exonuclease TaqMan Allelic Discrimination assay, which was performed utilizing minor groove binder probes fluorescently labeled with VIC or FAM and the protocol recommended by the supplier (Applied Biosystems, Foster City, CA, USA). Analysis for interpretation was performed with Via7 software and Taqman Genotyper software calls. Per

variant, concordance rate was obtained by comparing genotypes obtained from imputation and from TaqMan assays (Supplementary Table 6).

**Heritability estimation**. The genetic heritability based on genome-wide markers ($h_g^2$) was first estimated from the genetic relatedness matrix (GRM) after removing related individuals (–grm-cutoff 0.125) and corrected for population stratifications using the top 10 principal component (–qcovar), as implemented in GCTA[21,58]. Among a total of 14,044 enrolled HHCs, 692 progressed to active TB. Based on these numbers, we estimated the incidence rate in the Lima cohort for recent TB progression is 5%. Using this rate, we report $h_g^2$ on the liability scale to be 0.21 (s.e. = 0.08). If the true prevalence was in fact half as high, our estimate would instead be 0.17 (s.e. = 0.02); if twice as high, 0.26 (s.e. = 0.09). $h_g^2$ on the observed scale is 0.24 (s.e. = 0.09).

**Bayesian meta-analysis on GWAS summary statistics**. Briefly, we adopted a Bayesian meta-analysis approach[59] to test whether the reported top association is restricted to the early progressors only. We calculated the approximate Bayes factor (ABF)[28] for the top associated variant (rs73226617), testing the hypothesis that the reported association is specific to early progressors with a shared molecular fingerprint. We assumed the variance $\sigma^2$ around the true effect to be 0.04 as suggested by previous studies[28,60]. We assumed the probability of correlated true effects ($\rho$) between two phenotypes to be 0.5. The disease-specific $log_{10}(ABF)$(i.e., the ratio of the marginal likelihood for a model where the variant is only associated with early progressor who has a shared molecular fingerprints and/or a secondary cases ($log_{10}(ABF) = 5.81$) and for a model where is associated with all progressors ($log_{10}(ABF) = 6.19$) is −0.38. To test the robustness of the model using different priors ($\sigma^2$ and $\rho$), we varied the values of $\sigma = \{0.1,0.2,0.3,0.4\}$and $\rho = \{0,0.1,0.2,0.3,0.4,0.5,0.6,0.7,0.8,0.9\}$ but did not detect a strong difference that would alter the conclusion above (Supplementary Table 17).

**In silico functional annotation of candidate causal variants**. We combined multiple sources of in silico genome-wide functional annotations from publicly available databases to help identify potential functional variants and target genes in the 3q23 novel risk locus. To investigate functional elements enriched across the region encompassing the strongest candidate causal variants, we aggregated ~400 epigenomic and sequence annotations including cell-type-specific annotation types such as ATAC-seq, DNase-seq, FAIRE-seq, HiChIP-H3K27ac, HiChIP-CTCF, polymerase and elongation factor ChIP-seq, and histone modification ChIP-seq, as well as cell-type-nonspecific annotations, such as conservation scores and sequence annotation, such as coding, intronic, intergenic, etc. A list of all included resources is summarized in Supplementary Table 10, Dataset 3.

Using IMPACT[34], we built a model that predicts cell type gene regulatory elements by learning the epigenomic profiles of key TF binding sites in the cell type. Briefly, we trained IMPACT to distinguish regulatory elements from non-regulatory elements among 11 immune-related TFs and 4 others (Supplementary Table 18). To create the class of gold standard regulatory elements, we scanned the ChIP-seq peaks of the master TFs, mentioned above, for matches to the TF binding motif, using HOMER[61] [v4.8.3] and retained the genomic location of the highest scoring match for each ChIP-seq peak to the regulatory class. To create the class of non-regulatory elements, we scanned the entire genome for motif matches of each of the 14 master TFs, again using HOMER, and selected motif matches with no overlap with the ChIP-seq peaks. IMPACT learns an epigenomic profile representative of cell type regulatory elements in 10-fold cross validation (CV) using the complete sets of regulatory and non-regulatory elements. We scored regions of interest according to the learned feature profile from this CV.

**Electrophoretic mobility shift assay (EMSA)**. Frozen cell pellets from the THP1 cell line (ATCC) were used for preparation of nuclear extracts using NE-PER Nuclear and Cytoplasmic Extraction reagent (ThermoFisher) according to the manufacturer's instructions, then dialyzed overnight at 4 °C with gentle stirring in 1 L of pre-cooled dialysis buffer (10% glycerol, 10 mM Tris pH 7.5, 50 mM KCl, 200 mM NaCl, 1 mM di-thiothreitol, 1 mM phenylmethanesulfonyl fluoride). Samples were quantified using BCA Protein Assay Kit (ThermoFisher, catalogue no. 23227) and stored in 1× Halt protease inhibitor cocktail (ThermoFisher, catalogue no. 78437) at −80 °C until use. We designed single-stranded oligonucleotides (30–34 bp) corresponding to each set of alleles (Integrated DNA Technologies, Supplementary Table 19), and biotinylated the forward and reverse sequences separately using the Biotin 3'End DNA Labeling Kit (ThermoFisher Scientific, catalogue no. 20160) following the manufacturer's instructions. Single-stranded probes were annealed by incubation for 5 min at 95 °C followed by 1 h at room temperature. EMSA reactions were performed using the LightShift Chemi-luminescent EMSA kit (ThermoFisher, catalogue no. 20148). Binding reactions were performed in a volume of 20 µL: 2 µL of 10× binding buffer, 16 µg nuclear extract, 2.5% glycerol, 5 mM MgCl₂, 0.05% Nonidet P-40 and 50 ng Poly dI:dC as a nonspecific DNA competitor, and 20 mol of biotinylated probes with or without unlabeled competitor probes at 200 fold molar excess. The binding reaction was resolved on 5% Tris-borate-EDTA (TBE) gels (BioRad, catalogue no. 3450049) at 110 volts. Gels were transferred for 1 h at 4 °C at 40 volts onto pre-cut zeta-probe

nylon membrane (Bio-Rad, catalogue no. 162-0165). Transferred DNA was UV crosslinked for 10 min, then blocked and incubated with stabilize streptavidin-horseradish peroxidase conjugate, at 1:300 dilution in EMSA blocking buffer, then washed and detected by chemiluminescence. Finally, exposed on CL-Xposure™ films (ThermoFisher Scientific, catalogue no. 34089).

**CRISPR/Cas9 editing around the rs73226617 variant**. THP1 cells (ATCC, TIB-202) were cultured in complete RPMI (RPMI-1640, Gibco,10% fetal bovine serum, 1× non-essential amino acids, 15 mM HEPES, 2 mM L-glutamine, 1 mM sodium pyruvate, 0.05 2-mercaptoethanol, 1× penicillin-streptomycin). To disrupt the putative enhancer region around the rs73226617 lead variant, we selected 3 synthetic guide RNA (sgRNA, Synthego) molecules around rs73226617 (Supplementary Table 20) using Deskgen design tools (www.deskgen.com). We tested homology of the guide RNA sequence using the Basic Local Alignment Search Tool (BLAST) to confirm that the chosen sequences uniquely preferred the region around rs73226617 in 3q23, and did not have additional targets in the genome. For genomic editing, 40 μM total sgRNA and 40 μM of recombinant Cas9 protein (QB3 Microlabs) were mixed and incubated for 15 min at 37 °C to assemble CRISPR/Cas9 ribonuclear protein (RNP) complexes. Subsequently, $2 \times 10^5$–$10^6$ cells were nucleofected with 2 uL of RNPs in supplemented SG solution from the Cell line nucleofector X kit (V4XC-3032 SG, Lonza) using Amaxa 4D nucleofector (SG protocol: FF100). Bulk nucleofected cells were immediately topped with 37 °C prewarmed complete RPMI for 30 min, then cultured in a 24 well plate. RNA was extracted from bulk-edited samples 14–18 days after nucleofection. After 3 weeks, bulk-edited THP1 cells were single-cell sorted into four 96-well plates. The workflow is described in Supplementary Fig. 16.

To confirm the sequences of the edited genomic region, DNA was extracted from clonal THP1 cells using QuickExtract DNA Extraction Solution (Lucigen), and PCR-amplified with Q5 high fidelity DNA polymerase (New England Biolabs) using the protocol: 98 °C for 4', 35 cycles of (98 °C for 10", 65 °C for 30", 72 °C for 1'), 72 °C for 10', and 4 °C. PCR primers to amplify edited sequence: (Forward: TCTGGAAT TGAAGGGGCACA, and Reverse: AGCCCACCACACCTTTCTTT). Sizes and sequences of edited amplicons were verified by gel electrophoresis, and Sanger sequenced (GENEWIZ), respectively. Sequences were analyzed and aligned to the reference from unedited THP1 cells using SnapGene software (Supplementary Table 11).

**Gene expression analysis of edited THP1 cells**. RNA samples were extracted from each of the bulk-edited THP1 cells, as well as single-cell clones with RNeasy RNA isolation kit (Qiagen). RNA samples from expanded clones and three replicates of bulk-edited THP1 cells from three independent experiments, with matched Cas9 nucleofected cells without sgRNA were analyzed by low-input RNA sequencing (Genomics platform, Broad Institute). Libraries were sequenced using SmartSeq2 protocol.

For low-input RNA-seq, we used Kallisto version 0.43.1[62] to quantify gene expression using the Ensembl 83 annotation. We included protein-coding genes, pseudogenes, and lncRNA genes. We used log-transformed TPM (transcripts per million) as our main expression measure, which accounts for library size and gene size (specifically log2(TPM+1)). We considered as expressed genes those with a TPM > 1 in at least 95% of the samples. We further performed quantile normalization on the log2(TPM+1) values for our differential expression analyses. To test for differential expression, we fitted a linear regression model that included the first two principal components of gene expression as covariates.

**Reporting summary**. Further information on research design is available in the Nature Research Reporting Summary linked to this article.

## Data availability

Summary statistics is available through the NHGRI-EBI GWAS Catalog https://www.ebi.ac.uk/gwas/downloads/summary-statistics. Raw and processed RNA-sequencing data from the edited THP1 clones have been deposited in the GEO accession GSE134419. All scripts and data for generating figures presented in the manuscript are available at https://github.com/immunogenomics/TB_progression_GWAS. The source data underlying Figs. 2a and 3 and Supplementary Figs. 1, 3, 4, 5(c)–(d), 6, 7, 8, 10, 11, 12, 13, 14, 17 and 19 are provided as a Source Data file.

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

## Acknowledgements

The study was supported by the National Institutes of Health (NIH) TB Research Unit Network, Grant U19 AI111224 and NHGRI U01 HG009379. The content is solely the responsibility of the authors and does not necessarily represent the official views of the NIH. S.N. was supported by MRC (MR/M012328/1), the ERC Starting grant (260477) and the National Institute for Health Research (NIHR) Cambridge Biomedical Research Centre. T.A. was supported by NIH (NHGRI T32 HG002295). The authors thank Garðar Sveinbjornsson, Patrick Sulem, Ingileif Jonsdottir, and Kari Stefansson at deCODE genetics, Reykjavik, Iceland, for validating the association of rs73226617 with TB progression in the Icelandic population.

## Author contributions

Y.L. designed the genotyping array, performed statistical analysis of the GWAS data and wrote the first draft of the manuscript. S.S. performed the EMSA and CRISPR/Cas9 experiments. S.A. carried out the rare association studies of the GWAS data. T.A. implemented the IMPACT model. Y.B. helped to develop the protocols for the CRISPR/Cas9 experiments. K.I. performed the chromQTL analysis. M.G.A. helped the low-input RNA sequencing data analysis. R.C., L.L., S. R. L., J. J., R. Y., C.C., J. T. G., M.B. and M.B.M. participated in study design, protocol development, and sample collection. S.N. contributed the Russian data for the meta and heritability analysis. M.M.B. and P.A.N. helped to develop the protocols for EMSA experiments. D.B.M supervised the EMSA and CRISPR/Cas9 experiments. M.B.M. participated in study design, protocol development, and study conception. S.R. and M.B.M conceived and supervised the study. All authors contributed to the writing of the manuscript.

## Additional information

**Competing interests:** The authors declare no competing interests.

