## [Peer Review File · Nature Communications]

Reviewers' Comments:

Reviewer #1:

Remarks to the Author:

Luo et al. present an interesting paper describing human genetic susceptibility to early progression to active TB. The study is fairly well powered, especially for an ID GWAS, with approximately 4000 subjects, but what really sets it apart is the careful phenotype assignment. In contrast to previous GWAS of TB, this study using a longitudinal design, allowing for cases who were early progressors and controls to be exposed household contacts who did not develop active TB. This is a novel, labor-intensive, study design, so this manuscript could help inform design for future ID GWAS. The authors go on to make h^2 estimates, identify an apparent significant association, and attempt some *in silico* and *in vitro* functional validation. Unfortunately, while the authors are to be commended for their innovative design, the results are not particularly compelling or convincing and results are overstated.

Major:

- 1) The claim of greater h^2 in early progression compared to TB risk may be an overstatement. For early progression, the estimate was made with GCTA while for TB risk the estimate was made using a different dataset with LD score regression. What is the estimate for early progression based on LD score regression? In addition to the different methodologies for the 2 estimates, the populations are different as well— h^2 can be different between the populations for different reasons, including prevalence differences as described by the authors around line 330. Thus, this difference may not reflect an actual difference in the biology/genetic architecture of the traits. Finally, given the overlap of the estimates considering the SE's, is the claim even valid as stated? While the overall importance of the claim is debatable, the fact that the authors stressed this finding in the Abstract, requires that the claim be subjected to a greater level of scrutiny.
- 2) For similar reasons, the claim in line 135 of greater h^2 comparing 22.1 vs. 21.2% also seems dubious. Is the "larger" h^2 an important claim to make based on a <1% increase in h^2 with the given SE's in the estimates?
- 3) While replication is the gold-standard for GWAS studies, this threshold may be unreasonable given the lack of availability of such a unique dataset. The authors do conduct a second analysis, a stratified case analysis, which provides some additional validation, though I would have appreciated some discussion on how independent the results of this analysis should be considered.
- 4) It would be beneficial to provide better calibration of significance. $P < 5 \times 10^{-8}$ assumes 1 million independent tests. Phenotype permutation analysis would be useful to determine an empirical threshold for significance (as in Kanai et al 2016 J Human Genetics). Kanai et al. demonstrated with 1000 Genomes that for African populations this may not be stringent enough, while for admixed American populations it may be too stringent—so such an analysis could actually suggest greater confidence in the association given that it barely exceeds currently used $p < 5 \times 10^{-8}$ threshold.
- 5) In Figure 2, it appears that there is more admixture in the Controls vs. Cases based on the plots. Can the authors comment on that and how it may affect their conclusions?
- 6) The attempts to narrow down the functional SNP and perform functional followup are not convincing. Why IRF1 (vs. other IFN-responsive TFs or other TFs involved in inflammation and immunity) should be the focus of efforts here is not clear. The IMPACT analysis is done using macrophages, but the EMSA is done using Jurkat (a T cell line), and the luciferase reporters HEK293 (embryonic kidney). Why this multi-cell-type approach was taken is a bit puzzling. Transcriptional regulation and eQTLs can often be shared among different cell types, but they state that the IMPACT analysis is cell-type specific.
- 7) EMSA analysis. No indication of the number of biological replicates, quantification of signals, statistical analysis are given. Based on the amount of unbound sample, some of the differences may simply be due to unequal loading. Quantification of the EMSA signals from multiple experiments should help the investigators determine whether any of the signal is real. They may also want to rerun the IMPACT analysis based on T cells as the cell type instead of macrophages.

8) Luciferase assays show that none of the associated SNPs appear to affect expression in HEK293 cells. This should be repeated in the cell type where they have noted a difference in IRF1 occupancy—Jurkat and/or macrophage cells.

9) While a causal SNP is not convincingly identified, possible causal genes are given even less attention, thus limiting the impact of the manuscript in understanding TB pathogenesis.

Minor:

1) Typos throughout should be corrected, such as italicization of “exposure” in the title.

2) The novelty of conducting a genetic study of Peruvians may be overstated. A pubmed of “GWAS Peru” revealed several other studies that have incorporated Peruvian subjects. The authors should either scale back this claim or indicate more explicitly what differentiates this study from these previous studies.

3) Unclear what this sentence means at line 285: “Sex and age were included as fixed effects to correct for population stratification (Supplementary Figure 2).”

4) Overall, I think the manuscript would benefit if it weren't so compactly written.

Reviewer #3:

Remarks to the Author:

Luo and colleagues describe a GWAS of early TB progression in a Peruvian population. The study rationale, methods and results are clearly presented. The authors report a genetic locus at 3q23 as being associated with early TB progression. The authors highlight the relative paucity of validated infectious-disease genetic susceptibility loci, as compared to other complex traits, and advocate denser phenotyping as means to overcome the difficulty identifying infection-associated genetic variation.

The genetic association study design is excellent. The study participants are very well-phenotyped, which benefits the GWAS. The conduct and presentation of the GWAS itself is extremely robust, and I only have minor comments relating to that. However, the lack of independent replication of the GWAS findings makes the TB: 3q23 association interesting but preliminary. The particular design of the GWAS may well make replication in independent cohorts challenging, but in the absence of more convincing functional data, the genetic association needs to be replicated.

Major Points

1. The study does not include independent replication of TB progression susceptibility at the 3q23 locus. While I accept that the phenotype of TB progression would be challenging to replicate exactly, might it be possible to enrich for early progressors by restricting the Icelandic/Russian replication analysis to individuals under 40 years?

2. The authors state in the abstract that “early TB progression has a stronger genetic basis than population-wide TB susceptibility”. While I agree that the point estimate for heritability is higher for TB progression, the 95% confidence intervals for SNP heritability of TB progression and TB per se appear to overlap. A further limitation is that these estimates are derived by different means (were genotype level data not available for the Russian dataset?). The estimate of TB progression heritability merits reporting, but the interpretation needs to be more considered. It also merits some discussion that these estimates are sensitive to TB prevalence (as demonstrated in methods lines 329-331).

3. With regard to the case-only analysis it would be interesting to see the rs73226617 allele frequencies in clustered molecular fingerprint cases vs. unique molecular fingerprint cases vs. controls. Might it be informative to present a Bayesian analysis comparing progressors, reactivators vs shared controls, asking whether the most likely model is indeed an association restricted to the progressors?

4. The functional data highlights that the locus sits in a regulatory region in a plausible cell type, but does little to move forward our understanding of the biology underlying any TB association at 3q23. At a minimum, eQTL data supporting a cis association would be very helpful advancing our understanding of any disease association.

Minor Points

1. The sentence "We quantified.." on lines 97-98 seems redundant.
2. The study reports using SNP2HLA in the methods, but these results don't appear to be in the results. It would be of interest to report the associated classical HLA allele/amino acids at the class I locus.
3. Line 284-285: "Sex and age were included as fixed effects to correct for population stratification (Supplementary Figure 2)." The supplementary figure this refers to is presumably S3? Also I assume that this sentence is missing inclusion of principal components 1 and 2 (or was the GRM alone use to control for population structure)? Either seems appropriate, this just wasn't clear to me.
3. Line 241-2 methods – it reads as if cases in the main GWAS had to have an M.tb isolate shared with another case: "(2) index patients whose M. tb isolates shared a molecular fingerprint with isolates from other enrolled patients". But then why are there cases with unique M. tb molecular fingerprints in the case-only analysis?
4. The (negative) rare variant result should be reported in the body of the main text.

Overall summary statement

In looking over the thoughtful and carefully written reviews, we see that both reviewers are
largely in sync in the evaluation of our manuscript on human TB progression. Both reviewers
write highly positive and similar opinions with regard to the strengths of our submission as well
as specific suggestions for further validation. Specifically, they assess the core human subjects
design as being excellent, describing it as careful, novel, labor-intensive and robust. Beyond the
specific conclusions relating to a disease of worldwide importance, the reviewers acknowledge
that this careful study design and outcome could inform general approaches to infectious
disease GWAS in ways that overcome the difficulties in identifying infection-associated genetic
variation humans.

The reviews generally accept that the key claims outlined in the abstract relating to
population-specific genetic tools, quantitation of genetic heritability and identification of a novel
risk locus on 3q23, while also pushing for a secondary validation and more information about
candidate genes. We answer the latter issues in the point-by-point response and the revised
manuscript, which contains 13 new and revised figures, including

1. Statistical validations of main claims in the manuscript.
- ● **Supplementary Figure 6.** Heritability estimation using different methods (LDSC and
 - GCTA) for TB progression and population-wide TB susceptibility.
 - ● **Supplementary Figure 9.** Stratified cases with MAF of the top associated variant.
 - ● **Supplementary Figure 10-11.** Testing independence between the primary the
 - secondary association studies.
 - ● **Supplementary Figure 12.** New association analysis of the HLA region.
 - ● **Supplementary Figure 19.** Genome-wide association study with Native American
 - ancestry proportion as an additional covariate in the linear mixed model.
- 2. In silico search for existing evidence of promoter/enhancer activities that suggest
ATP1B3 as a potential causal gene highlighted by the identified novel risk locus.
- ● **Figure 3e.** Predicted cell-state regulatory activity among each variant in 15 different cell
 - types and cell states using IMPACT.

- • **Supplementary Figure 15.** Promoter activity suggested by Promoter capture Hi-C.
• **Supplementary Figure 16.** chromQTL signals identified in the Blueprint project.
• **Supplementary Figure 18.** Enhancer activity in primary monocytes and THP1 cell lines
suggested by ChIP-seq and ATAC-seq.
3. New functional validations.
• **Supplementary Figure 14.** Updated EMSA experiment in THP1 cell line.
• **Supplementary Figure 16-17.** Overview of the CRISPR/Cas9 experiment and
differential expression test.

Specific comments

Reviewer #1 (Remarks to the Author):

Luo et al. present an interesting paper describing human genetic
susceptibility to early progression to active TB. The study is fairly
well powered, especially for an ID GWAS, with approximately 4000
subjects, but what really sets it apart is the careful phenotype
assignment. In contrast to previous GWAS of TB, this study using a
longitudinal design, allowing for cases who were early progressors and
controls to be exposed household contacts who did not develop active
TB. This is a novel, labor-intensive, study design, so this manuscript
could help inform design for future ID GWAS. The authors go on to make
h² estimates, identify an apparent significant association, and
attempt some in silico and in vitro functional validation.
Unfortunately, while the authors are to be commended for their
innovative design, the results are not particularly compelling or
convincing and results are overstated.

**Response:** We appreciate the assessment that our unique study design and its potentially
broad impact that could change general approaches to infectious disease genetic studies, which
have particular complexities related to host and pathogen interactions.

Major:

1) The claim of greater h^2 in early progression compared to TB risk
may be an overstatement. For early progression, the estimate was made
with GCTA while for TB risk the estimate was made using a different
dataset with LD score regression. What is the estimate for early
progression based on LD score regression? In addition to the different
methodologies for the 2 estimates, the populations are different as
well— h^2 can be different between the populations for different
reasons, including prevalence differences as described by the authors
around line 330. Thus, this difference may not reflect an actual
difference in the biology/genetic architecture of the traits. Finally,
given the overlap of the estimates considering the SE's, is the claim
even valid as stated? While the overall importance of the claim is
debatable, the fact that the authors stressed this finding in the
Abstract, requires that the claim be subjected to a greater level of
scrutiny.

Response:

We thank the Reviewer for pointing out that the different methods for estimating genetic
heritability may have accounted for the differences between the reported heritability of
progression compared to general TB susceptibility. We noted that LDSC is not suitable for
admixed populations, we therefore requested access to the raw imputed genotype data from the
Russian cohort and applied the same GCTA analysis that we used to analyze the data from the
Peruvian cohort. The new analysis obtained a h_g^2 estimates of 0.178 ± 0.02 (compared to
0.155 ± 0.04 using LDSC). This is, indeed, not statistically different ($p=0.68$) from the h_g^2 for
early TB progression (0.212 ± 0.08). We have now **removed** the claim that TB progression has
a stronger genetic basis than population-wide TB susceptibility from the abstract:

“Compared to the reported h_g^2 of genome-wide TB susceptibility (15.5%), this result
indicates early TB progression has a stronger genetic basis than population-wide TB
susceptibility.”

And replaced it with:

“This degree of heritability suggests TB progression has a strong genetic basis, and is
comparable to traits with well-established genetic bases.”

We also added the standard error (0.08) when reporting h_g^2 . We then edited the following text
to describe these results in the main text in the heritability section:

“To compare the genetic heritability between early TB progression and population-wide
TB susceptibility, we subsequently obtained genotypes from a previous TB study
conducted in Russia with 11,137 individuals¹. Using GCTA, we estimated the h_g^2 of
population-wide TB susceptibility to be 17.8% (s.e.=0.02, $P = 2.85 \times 10^{-21}$) with
assumed prevalence of 0.04². Even though the point estimate of h_g^2 of TB progression is
greater than that of population-wide TB risk in the Russian study, these estimates are not
statistically different from each other (two tailed $P = 0.68$, **Supplementary Figure 6**).
Regardless, the strong host genetic basis of TB progression suggests that larger
progression studies may be well-powered to discover additional variants.”

In addition, we added a new **Supplementary Figure 6**, to summarize and help readers better
understand these heritability estimates:

**Supplementary Figure 6. Heritability estimates of TB progression and population-wide TB**
**susceptibility.** Each bar plot represents the genetic heritability estimates (h_g^2) based on
different cohort definition and statistical method that had been employed as described in the
x-axis. The number of samples used in each estimation is reported in the bracket.

2) For similar reasons, the claim in line 135 of greater h^2 comparing
22.1 vs. 21.2% also seems dubious. Is the "larger" h^2 an important
claim to make based on a <1% increase in h^2 with the given SE's in the
estimates?

**Response:**

We thank the Reviewer for pointing out the statistically non-significant differences ($p=0.93$)
between these two point estimates. We **removed** the claim of a 'larger', and only reported the
statistics of h_g^2 estimates. As per **Comment #1**, we added a new **Supplementary Figure 6** to
address this concern.

3) While replication is the gold-standard for GWAS studies, this
threshold may be unreasonable given the lack of availability of such a

unique dataset. The authors do conduct a second analysis, a stratified
case analysis, which provides some additional validation, though I
would have appreciated some discussion on how independent the results
of this analysis should be considered.

**Response:**

We agree regarding the need for validation and that an independent test would increase the
credibility of our stratified case analysis. In revision, we further investigated whether there is a
correlated effect size between case-control and case-only analysis. If these association studies
are statistically dependent, then we would expect a correlation between reported effect sizes.
Instead, the correlation between these two analyses is negligible (Pearson correlation = 0.014,
new **Supplementary Figure 10**), suggesting an independent relationship.

Next, to rigorously assess the significance of our reported effect size, we conducted a
permutation test within the case-only analysis. Here, we randomly permute the within-case
status (early progressor versus others) 10,000 times. This analysis concludes that the observed
OR of 1.09 has a P-value of 0.017 (new **Supplementary Figure 11**) compared to null. These
results confirm that our secondary analysis is independent from the primary analysis and have a
nominal association with early progression after recent exposure to *M.tb*.

We have added the text to reflect this independence of our stratified case analysis:

“To assess the independence of the stratified cases compared to the overall
case-control analysis, we first compared reported effect sizes in both analyses and
observed a low Pearson correlation ($r = 0.014$, **Supplementary Figure 10**). To test the
significance of the reported association, we performed a permutation analysis, where we
randomly permuted the case/control status in the stratified analysis. After permuting for
10,000 times, the observed OR (1.09) has a P-value of 0.017 (**Supplementary Figure**
**11**). “

In addition, we have added two new **Supplementary Figures 10** and **11** to address this
concern.

**Supplementary Figure 10. Correlation between effect size (beta) between case-control**
**(active TB cases versus latent TB controls) analysis and within case (early progressors**
**versus other TB cases).** Each dot in the plot represents a genetic variant, if the two tests are
dependent, then there should be a non-zero correlation between two betas. Instead, we
observed a Pearson correlation of 0.014, suggesting the secondary, within case-only, analysis
can be considered as independent test compared to the primary (case-control) analysis. The
SNP (rs73226617) highlighted in red is the top associated risk variant.

**Supplementary Figure 11. Random permutation test of individuals in early and other**
**progressors among active TB cases.** The distribution of effect size was generated by
randomly assigning early and other status among 2,160 TB cases. The red line in the panel
marks the actual effect size observed. We conclude the observed OR of 1.09 has a P-value of
0.017 compared to null.

4) It would be beneficial to provide better calibration of
significance. $P < 5 \times 10^{-8}$ assumes 1 million independent tests. Phenotype
permutation analysis would be useful to determine an empirical
threshold for significance (as in Kanai et al 2016 J Human Genetics).
Kanai et al. demonstrated with 1000 Genomes that for African
populations this may not be stringent enough, while for admixed
American populations it may be too stringent—so such an analysis could

actually suggest greater confidence in the association given that it
barely exceeds currently used $p < 5 \times 10^{-8}$ threshold.

**Response:**

We thank the Reviewer for pointing out the differences in the statistical thresholds for
significance of associations in the GWAS study, as well as the useful suggestions for an
alternative/more appropriate genome-wide significance threshold for admixed population.
Following the same permutation strategy as presented in *Kanai et al. 2016* for the 85 Peruvian
individuals included in the 1000 Genomes Project, we estimated the empirical genome-wide
significance threshold in this population group to be 1.78×10^{-7} assuming ~9.6 million variants
with MAF $\geq 1\%$. Our top associated signal rs73226617 ($P = 3.93 \times 10^{-8}$) becomes more
significantly associated with TB progression. We subsequently added the more appropriate
significant threshold in **Figure 3** and added the additional reference. We also replaced the
original $P < 5 \times 10^{-8}$ in the abstract to $P = 3.93 \times 10^{-8}$, so that the readers know the exact
P-value in the association test. We have also added the following line to the methods:

"To determine an appropriate genome-wide significant threshold for Peruvian
populations, we followed the permutation strategy proposed by *Kanai et al. 2016*³, and
considered a variant is significantly associated with TB progression, if it has a P-value
smaller than 1.78×10^{-7} ."

We further updated **Figure 3a** to incorporate this new genome-wide significant threshold:

**Figure 3. Genome-wide association details of the 3q23 locus.** (a) A regional association plot
of the 3q23 locus including all genotyped and imputed variants. The horizontal line indicates the
genome-wide significant threshold at \$1.78 \times 10^{-7}\$ for Peruvian populations³.

5) In Figure 2, it appears that there is more admixture in the
Controls vs. Cases based on the plots. Can the authors comment on that
and how it may affect their conclusions?

**Response:**

We thank the Reviewer for having noticed the ancestry differences between cases and controls.
In the original linear mixed model presented, we included only sex and age as fixed effects. To
test whether the differences in admixture percentage between cases and controls affect our
genome-wide association studies, we included the inferred Native American percentage from
the ADMIXTURE analysis (K=6) as an additional covariate in revision. Overall, inclusion of the
Native American ancestry proportion as an additional covariate in the model did not affect the
main conclusions (new **Supplementary Figure 19**). We further reported the association results
of the risk locus after condition on their ancestral proportion in the updated **Supplementary**
**Table 6**.

We added the following text in the Method section to specifically address this concern:

“To control for the potential effect of ancestry differences between cases and controls
and the robustness of our reported findings, we tested our linear mixed model adding
Native American ancestry inferred from ADMIXTURE analysis (K=6) as a covariate. We
observed similar association strengths genome-wide (**Supplementary Figure 19**) and in
our reported top associations (**Supplementary Table 6**). This result supports that our
reported associations are independent of individual ancestral proportions.”

We added the following new **Supplementary Figure 19** to support our claim:

**Supplementary Figure 19. Manhattan and QQ-plots of TB progression including the**
**Native American proportions as a covariate in the linear mixed model. Manhattan (top) and**

QQ (bottom) plot showing genome-wide association study for single common variants
(6,035,269, MAF \geq 1%). P-values were reported from the linear mixed model using the genetic
relatedness matrix (GRM) as random effects. Sex, age and Native American proportions
inferred from the ADMIXTURE analysis (K=6) were included as fixed effects. The diagonal black
line in all QQ-plots is $y = x$, and the grey shapes show 95% confidence interval under the null. λ
240 s are the genome-wide inflation factors based on all tested statistics.

Detailed statistics of the 11 top associated variants were also reported in **Supplementary Table**
**6**:

rsID	effect size	standard error	P-value
rs73239724	0.149	0.031	2.08E-06
rs73226608	0.154	0.031	8.78E-07
rs58538713	0.162	0.031	1.78E-07
rs11710569	0.155	0.031	8.30E-07
rs11714221	0.149	0.031	1.68E-06
rs189348793	0.155	0.031	7.90E-07
rs73226617	0.166	0.030	3.65E-08
rs148722713	0.156	0.031	6.65E-07
rs73226619	0.151	0.031	1.34E-06
rs112304167	0.166	0.034	1.01E-06
rs146526750	0.169	0.035	1.32E-06

We are separately interested in whether ancestry differences for individuals overall
genome-wide may be associated with TB progression. There is some precedent for this in the
literature⁴⁻⁸. However, these differences can be confounded by socioeconomic and other
economic factors causing subtle stratification. We are now looking at this specific issue in a
separate and detailed study.

6) The attempts to narrow down the functional SNP and perform
functional followup are not convincing. Why IRF1 (vs. other
IFN-responsive TFs or other TFs involved in inflammation and immunity)
should be the focus of efforts here is not clear. The IMPACT analysis

is done using macrophages, but the EMSA is done using Jurkat (a T cell
line), and the luciferase reporters HEK293 (embryonic kidney). Why
this multi-cell-type approach was taken is a bit puzzling.
Transcriptional regulation and eQTLs can often be shared among
different cell types, but they state that the IMPACT analysis is
cell-type specific.

**Response:**

We acknowledge the Reviewer's concern about the differences in cell types used for functional
validation. This is a previously uncharacterized locus, and therefore, it was unclear what the
most relevant cell type and context choice should be. We therefore conducted more thorough
and deeper analyses to get at this issue.

We first looked for cell-type-specific regulatory elements using an updated version of IMPACT⁹
and observed monocyte-specific predicted regulatory elements at rs73226617 at
chr3:141400653 and rs148722713 at chr3:141401146 (IMPACT score 0.79 and 0.41
respectively). We recently demonstrated that IMPACT is able to outperform predictions of
cell-type specific transcription factor binding better than other epigenetic features, or indeed
other motif prediction algorithms⁹.

Based on the IMPACT analysis and the suggested enhancer activity in monocytes, we studied
monocytic cell lines (THP1) as the most likely experimental model for locus-specific gene
regulatory activity, recognizing that THP1 immortalized cell lines may only approximate the
biology of monocytes. In order to consolidate the analyses, we performed the EMSA analysis in
human THP1 cells. We have revised our EMSA analysis so that it is now focused on
allele-specific binding of THP1 nuclear complexes, and used different types of DNA retardation
gels to address technical concerns (see response to **Comment #7**). We have removed data
from Jurkat76 cell lines (representing T cells).

To demonstrate allele specific enhancer activity, we carried out extensive luciferase experiments
in THP1 cells, but could not fully implement the technique due to the lower transfection
efficiency, which renders the quantitative assessment of allele-specific effects on luciferase
reporter expression unreliable. We subsequently removed the luciferase analysis from the

manuscript completely to harmonize all the laboratory validations in monocytes, as predicted by
the *in silico* analysis (see response to **Comment #8**).

We addressed the choice of using different cell lines in the EMSA concerns more specifically in
the following text:

“Briefly, IMPACT identifies regions predicted to be involved in transcriptional regulatory
processes related to a cell-type-specific key transcription factor (**Methods**) by leveraging
information from approximately 400 chromatin and sequence annotations in public
databases (**Figure 3c, Supplementary Table 10**). Each variant is assigned with a
probability between 0 (least likely to be a regulatory element) and 1 (most likely to be a
regulatory element). We tiled through the 23,308 base pair region on a per-nucleotide
basis, computing the probability of a cell-type-specific regulatory element separately for
15 different cell types and cell states of which 10 are immune cell types with known roles
in TB outcomes, including T cells, B cells, monocytes, macrophages, and peripheral
blood cells (**Figure 3e**). We observed monocyte-specific predicted regulatory elements
at rs73226617 and rs148722713 (IMPACT score 0.79 and 0.41 respectively, **Figure 3d**).

...

Based on the IMPACT analysis and the suggested enhancer activity in monocytes, we
studied monocytic cells (THP1) as the most likely experimental model for locus-specific
gene regulatory activity. We performed electrophoretic mobility shift assays (EMSA) to
test whether the variants differentially bound nuclear complexes in an allele-specific
manner among the seven variants that constitute the 90% credible set (**Methods**). We
could detect differential protein binding that was competed out by unlabeled probes for
three of the risk alleles (rs73226617, rs58538713, and rs148722713) (**Supplementary**
**Figure 14**), providing evidence that these alleles might confer differential transcription
factor binding activity, and in the right context may lead to altered enhancer activity.”

For the new IMPACT analysis we added the following new **Figure 3e**:

**Figure 3e.** Intersection of nucleotide-resolution of variant cell-state IMPACT annotations with
 potential causal variants in 3q23 locus. The y-axis shows the posterior probability of predicted
 cell-state regulatory activity among each variant in 15 different cell types and cell states. The
 x-axis shows the genomic positions of all 11 risk variants among the identified risk locus. The
 bolded variant (rs73226617) is the leading risk variant from the association study which shows
 the highest predicted cell-state regulatory activity in monocytes (masked by CEBPB
 transcription factor).

We have also updated main **Figure 3c-d** in light of the new IMPACT analysis

**Figure 3.** (c) Number of overlaps between all variants in the risk locus and ~400 epigenetic
features. (d) Predicted posterior probability of cell-type-specific gene regulatory activity using
Inference and Modeling of Phenotype-related ACtive Transcription (IMPACT) based on the
epigenetic chromatin signature of binding sites of the transcription factor CEBPB in monocytes.
Dashed lines highlights 11 top associated variants. Genotyped variant rs73226617 is highlighted
in red bar.

7) EMSA analysis. No indication of the number of biological
replicates, quantification of signals, statistical analysis are given.
Based on the amount of unbound sample, some of the differences may
simply be due to unequal loading. Quantification of the EMSA signals
from multiple experiments should help the investigators determine
whether any of the signal is real. They may also want to rerun the
IMPACT analysis based on T cells as the cell type instead of
macrophages.

**Response:**

The EMSA analysis is now reported in THP1 cell lines, instead of Jurkat76 cells

**(Supplementary Figure 14)** In terms of replication, we performed the EMSA in THP1 nuclear

extract samples derived from three independent cell line batches, which showed consistent
patterns of probe binding to nuclear complexes.

In terms of quantification and statistical analysis of the EMSA results, we did not statistically
evaluate broad patterns in allele-specific binding to THP1 nuclear complexes, since we interpret
the EMSA results qualitatively, to evaluate broad patterns in allele-specific differential binding of
probes to THP1 nuclear complexes. They only lend one layer of evidence to functionally
validate allele-specific binding activities, and are used in this context as an initial screen to
prioritize variants for functional follow-up without deriving any mechanistic or quantitative insight.

Finally, the Reviewer raises a specific concern about the amount of unbound “free probe” in the
previous EMSA analysis, which we address in revision. We realized that the previous EMSA
gradient (6-12%) gels showed an artifact where the signal in the unbound free probe at the
bottom of the gel was lost when the non-biotinylated “cold” competitor probes were added. As
the Reviewer correctly points out, the loss of signal when the competitor was added would cloud
the interpretation of allele-specific binding patterns, since it would not be clear if the signal was
lost because of this gel type-associated artifact or a real competition between the biotinylated
and non-biotinylated probes. We re-ran the EMSA experiments using a 5% Tris-base-EDTA
(TBE) gels, which showed the unbound free probe at the bottom more clearly, as well as more
equivalent loading in the different wells. In these cases, comparing the second lane (biotinylated
probe only) and the third lane (biotinylated and non-biotinylated probes) for each allele still
showed equivalent amounts of free probes at the bottom of the gels, suggesting that
competition did not result in a loss of the biotinylated probe signal.

As per previous **Comment #6**, we reperformed the IMPACT analysis in 15 different cell types
and cell states, and saw little signal in T cells and other cell types at our SNPs with highest
posterior probabilities. To focus on the monocyte lineage, we reported the EMSA analysis using
THP1 cell lines in the manuscript to address this point.

We also updated **Supplementary Figure 14** for the EMSA results performed in THP1, which
also address the concern about the disappearance of the unbound “free probe” after we used
the 5% TBE DNA retardation gels. Here is an example:

**Supplementary Figure 14. EMSA for top seven associated variants.** (a) rs73226617 (b)
rs58538713 (c) rs148722713 (d) rs189348793 (e) rs11710569 (f) rs73226608 (g) rs146526750.
Lanes in the panel correspond to double stranded probes without (lane 1) or with THP1 nuclear
extracts (lane2) and an additional non-biotinylated competitor probe (lane 3).

8) Luciferase assays show that none of the associated SNPs appear to
affect expression in HEK293 cells. This should be repeated in the cell
type where they have noted a difference in IRF1 occupancy—Jurkat
and/or macrophage cells.

**Response:**

We acknowledge the Reviewer's concern about the various cell lines used for functional
validation of the 3q23 variants. We removed the luciferase analysis from the manuscript
completely to harmonize all the laboratory validations in monocytes, as predicted by the *in silico*
analysis.

Similarly to the EMSA, we attempted to use THP1 cells for this experiment. However, due to the
low lipofectamine-based transfection efficiency of THP1 cells as shown in the luminescence
signals in the different transfected cell lines in attached figure, we opted to use human
embryonic kidney (HEK) 293T cells, which are routinely used for luciferase assays. The
luminescence readout for both the Firefly and Renilla luciferase vectors was more than 2 logs

higher in HEK293T cells than either Jurkat 76 or THP1 cells (see below), and were more
reproducible across technical replicates. Therefore, we considered the luminescence data more
reliable in HEK293T cells, compared to THP1 cells.

Overall changing the cell line alone for this assay may not be sufficient to unmask false negative
findings. We conclude that failure to detect allele-specific functional activities using luciferase
assays does not preclude a cell type- and context-specific gene regulatory activity in the locus.

9) While a causal SNP is not convincingly identified, possible causal
genes are given even less attention, thus limiting the impact of the
manuscript in understanding TB pathogenesis.

**Response:**

To address the Reviewer's concern, we first performed a number of *in silico* lookups (promoter
Hi-C, eQTL and chromQTL) for existing evidence of promoter/enhancer activity that can suggest
potential causal genes highlighted by the identified novel risk locus.

We conducted a new set of experiments using CRISPR/Cas9 to introduce indels to disrupt the
3q23 enhancer region where the candidate variants are concentrated (**Supplementary Figures**
**16 and 17**). Due to the well-documented difficulties of CRISPR/Cas9 editing primary human
monocytes, we investigated these loci in THP1 cells, a well-studied monocytic cell line. We
hypothesized that disrupting the putative enhancer region would modulate the expression of
neighboring genes, thus pointing to the most likely gene associated with the risk allele. We
generated individual THP1 clonal cell lines harbouring unique edits and deletions in the proximal
region surrounding rs73226617 using target guides and compared gene expression between
edited and unedited clonal cell lines. However, we could not detect any differential gene
expression as a consequence of disrupting the putative enhancer region. Therefore, we could
not definitely conclude that the region regulates the expression of any proximal or distal genes,
under the tested cell type and context.

We added the following text to describe the new *in silico* evidence for potential causal gene
candidate in the main text:

"Next we searched public promoter Hi-C databases^{10,11} to identify any significant
interactions between the monocyte specific enhancer harboring our most likely causal
allele, rs73226617 and rs148722713. We found that in monocytes, both of the risk
variants (rs73226617, rs148722713) are in a region that interacts with the promoter of
*ATP1B3* (**Supplementary Figure 15a-b**). Similar to the IMPACT results, we found the
variant-gene interactions are strongest in monocytes compared to other cell types
(**Supplementary Figure 15c-d**), suggesting cell-type-specific activities in the identified
TB risk locus. *ATP1B3* (ATPase Na⁺/K⁺ Transporting Subunit Beta 3) is a protein-coding
gene, which belongs to the family of Na⁺/K⁺ and H⁺/K⁺ ATPases. Na⁺/K⁺ -ATPases are

composed of an alpha, beta, and FXYD subunits, are integral membrane proteins
responsible for establishing and maintaining the electrochemical gradients of sodium
and potassium ions across the plasma membrane through active transport against their
osmotic gradients. A recent study demonstrated that the Na, K ATPase Beta 3 subunit in
monocytes has an important function in mediating a normal T cell response¹². Indeed
ligating it with an antibody resulted in a blunted T cell response after stimulation. This
effect was specific to the monocytes population. Consistent with these findings,
differential expression of *ATP1B3* in whole blood, along with genes coding for other
members of the Na⁺/K⁺ -ATPases, was recently reported to be associated with TB
progression in an African cohort of household contacts of TB patients¹³. Collectively, the
Hi-C analysis and reported association with TB progression point to *ATP1B3* as a
candidate gene of the risk locus in 3q23. ”

We added new **Supplementary Figure 15** to report the promoter activity supported by Hi-C
data:

**Supplementary Figure 15. Promoter capture Hi-C from www.chicp.org.** Selected public
 promoter Hi-C data in 17 human primary hematopoietic cell types reveals (a)-(b) strong
 monocyte interactions (highest score = 9.54) between an enhancer region containing the
 leading risk variant (rs73226617) and *ATP1B3* in monocyte. This interaction is much weaker in
 (c)-(d) the Naive CD4+ T cells and other cell types (highest score = 5.51).

We added new **Supplementary Figure 13** to report the enhancer activity supported by
 chromQTL data:

**Supplementary Figure 13. Chromatin QTL analysis results in Blueprint project.** To
 understand the effects of genetic variants in immune cells, we utilized eQTL
 summary statistics produced by Blueprint project¹⁴. Detailed methods were reported in the
 original article. Briefly, the Blueprint project collected CD14+ monocytes (brown), CD16+
 neutrophils (grey), and naive CD4+ T cells (light blue) from 197 individuals. We analyzed
 histone variation (H3K4me1) and tested associations of genetic variants within 1 Mb of each
 normalized features using a linear regression model that includes a random effect term
 accounting for sample relatedness. Four top risk variants that are associated with TB
 progression were included in the analysis (annotated in white boxes).

We added the following text to describe our new CRISPR/Cas9 experiment in the manuscript:

“Since *in silico* evidence suggested that our identified TB risk locus harbors
 monocyte-specific regulatory elements, we used the CRISPR/Cas9 system to introduce
 insertions/deletions around the top associated variant rs73226617 (**Methods**,
 **Supplementary Figure 16a**). Among 23 sorted and grown clones that had unchanged
 risk loci or harbored unique edits and deletions (**Supplementary Table 11** and
 **Supplementary Figures 16b-c**), we did not observe differential gene expression
 between edited and unedited THP1 clones in the eight genes around the rs73226617

variant ($P > 0.05$, **Methods, Supplementary Figures 17**). While we observed no effect in
 THP1 cell lines, this might be the result from differences between primary monocytes
 and transformed THP1 cell lines, or failure to identify the relevant activation conditions
 and cell context to test enhancer activities, which are known to influence eQTL
 interactions^{15–18}. In particular we noted the enhancer activity seen in primary monocytes,
 is not seen in THP1 cell lines^{19–22} (**Supplementary Figure 18**).”

We added **Supplementary Figure 16** and **17** for details of the CRISPR/Cas9 experiment:

**Supplementary Figure 16. Overview of the CRISPR/Cas9 experiment.** (a) CRISPR/Cas9
 strategy to disrupt the enhancer region surrounding the rs73226617 lead risk variant in 3q23.
 THP1 cells were nucleofected with 3 guide RNA molecules targeting genomic region around the
 variant, then expanded for RNA extractions and gene expression analysis. Bulk-edited THP1
 cells were also single-cell sorted into 96 well-plates and expanded for DNA extractions and
 sanger sequencing for initial screening. 23 clones were expanded to represent different edits,
 where some show evidence of genomic deletion, or intact sequence length, for gene expression
 analysis by low-input RNA sequencing and qRT-PCR. (b) Amplicons were analyzed by gel
 electrophoresis to confirm deletions detected after initial screening. Intact amplicons are
 expected around 700 base pairs (wildtype band, far left). (c) Alignment of sanger sequences

derived from the 23 THP1 clones showing location of edits compared to wildtype (unedited)
amplicon sequences. Red and blue sequences represent edited and unedited THP1 clones,
respectively.

Supplementary Figure 17. Low-input RNA-sequencing analysis. Expression of eight genes
around rs73226617 in THP1 clones, which maintained wildtype genomic sequence after
expansion of single cells from bulk-edited THP1 cells compared to edited clones. P-values are
derived from a linear regression model including first principal component of the gene
expression profile as covariate.

We added **Supplementary Figure 18** to show the enhancer activity was only seen in primary
monocytes.

**Supplementary Figure 18. Enhancer activity of the risk locus (3q23) in primary**
 **monocytes and THP1 cell lines indicated by ChIP-seq and ATAC-seq.** From top to bottom,
 the y-axis shows the raw reads of ChIP-seq for H3K4me1 in primary monocytes (GSM1003535)
 and in THP-1 cell lines (GSM3514950); raw counts of ATAC-seq in primary monocytes
 (GSE74912) and in THP1 cell lines (GSE96800). The x-axis shows the genomic positions of the
 identified risk locus (chr3:141383525-141407033). The vertical lines highlights 11 top
 associated variants. Genotyped variant rs73226617 is highlighted in bold.

Minor:

1) Typos throughout should be corrected, such as italicization of
 "exposure" in the title.

**Reponse:**

We have corrected typographical errors.

2) The novelty of conducting a genetic study of Peruvians may be
overstated. A pubmed of "GWAS Peru" revealed several other studies
that have incorporated Peruvian subjects. The authors should either
scale back this claim or indicate more explicitly what differentiates
this study from these previous studies.

**Reponse:**

Our claim that our study 'is the most extensive genetic study conducted in Peru to date' reflect
the fact that, to our knowledge, the number of subjects enrolled in our study (4,002) is the
largest that has been conducted in Peru to date. We had another search of public database of
Peruvian genome in the published work, the next largest study that has been recently published
included 1,247 Peruvian subjects²³. Based on the Reviewer's recommendation, we have thus
revised the text:

"To our knowledge, this represents the largest genetic study conducted in Peru to date."

3) Unclear what this sentence means at line 285: "Sex and age were
included as fixed effects to correct for population stratification
(Supplementary Figure 2)."

**Response:**

We have now changed the text:

"We used the genetic relatedness matrix (GRM) as random effects to correct for cryptic
relatedness and population stratification between collected individuals. Sex and age
were included as fixed effects."

4) Overall, I think the manuscript would benefit if it weren't so
compactly written.

**Response:**

We thank the Reviewer for this suggestion and have expanded our text where possible,
especially around efforts on functional validations and biological implications of our study.

Reviewer #3 (Remarks to the Author):

Luo and colleagues describe a GWAS of early TB progression in a
Peruvian population. The study rationale, methods and results are
clearly presented. The authors report a genetic locus at 3q23 as being
associated with early TB progression. The authors highlight the
relative paucity of validated infectious-disease genetic
susceptibility loci, as compared to other complex traits, and advocate
denser phenotyping as means to overcome the difficulty identifying
infection-associated genetic variation.

The genetic association study design is excellent. The study
participants are very well-phenotyped, which benefits the GWAS. The
conduct and presentation of the GWAS itself is extremely robust, and I
only have minor comments relating to that. However, the lack of
independent replication of the GWAS findings makes the TB:3q23
association interesting but preliminary. The particular design of the
GWAS may well make replication in independent cohorts challenging, but
in the absence of more convincing functional data, the genetic
association needs to be replicated.

**Response:** We are grateful for the strong endorsement of our key findings as excellent, since
the basic lack of understanding the genetic basis of why some humans progress to TB, but most
do not, remains one of the most important unanswered questions in the TB field. We accept that
independent population validation is important, so have revised on key issues as highlighted
above and a few additional points that are framed according to the Reviewer's major points.

Major Points

1. The study does not include independent replication of TB
progression susceptibility at the 3q23 locus. While I accept that the
phenotype of TB progression would be challenging to replicate exactly,
might it be possible to enrich for early progressors by restricting

the Icelandic/Russian replication analysis to individuals under 40
590 years?

**Response:**

We agree with the Reviewer's comment recognizing the challenge of replicating the study in an
independent validation cohort due to the uniqueness of our cohort definition. Having said that,
we were also uncertain whether restricting the age of Icelandic/Russian patients to those under
40 would necessarily obtain rapid progressors. In both instances the controls are uninfected
individuals.

In order to address the Reviewer's concern we pursued the suggestion of a possible replication
strategy by restricting cases in the Icelandic/Russian cohort to individuals under 40 years old, to
increase the likelihood of primary progression, thus better resembling the progressors in the
Peruvian cohort. We subsequently contacted authors in the Russian cohort and obtained age
information in cases. In the association study where we restrict cases with individual under 40
604 years old only, we observed a P-value of 0.673 that is as compared to 0.065 in the non-stratified
association study. We noted this result is opposite to what we expected for early progressors (as
summarized in the figure below). We speculated that since the incarcerated population
accounted for ~25% of all new TB cases in Russia, and the majority of TB cases among
prisoners were identified during the initial examination²⁴, suggesting that many cases had been
missed by the civilian TB centers and thus age at diagnosis might not be a good indicator for
early progression in this cohort.

Further, we noted that the frequency of the top risk variant (rs73226617) in the 'non-rapid'
progressor population is similar to the frequency reported in the general population (3%,
reported in the 1000 Genome Project). However, its frequency is lower among latent controls
(MAF = 2.1%), and higher in early progressors (MAF = 4.2%), suggesting that significant
association that we observed is contributed by both the early progressors and latent TB
individuals. This is confirmed by the frequency that we observed in the Russian cohort for the
same risk variant has the same frequency in controls as it is reported in the general population
(MAF = 8%). Therefore, the differences in the phenotypic definition for the control samples could
significantly lower the power for detecting the same association in a population-wide TB
susceptibility cohort.

2. The authors state in the abstract that “early TB progression has a
 stronger genetic basis than population-wide TB susceptibility”. While
 I agree that the point estimate for heritability is higher for TB
 progression, the 95% confidence intervals for SNP heritability of TB
 progression and TB per se appear to overlap. A further limitation is
 that these estimates are derived by different means (were genotype
 level data not available for the Russian dataset?). The estimate of TB
 progression heritability merits reporting, but the interpretation
 needs to be more considered. It also merits some discussion that these
 estimates are sensitive to TB prevalence (as demonstrated in methods
 lines 329–331).

**Response:**

We agree that comparing heritability estimates derived using different methods are not
 straightforward. We requested genotype level data from the Russian dataset, and performed
 GCTA analysis for estimating the genetic heritability of population-wide TB susceptibility. As per
 **Reviewer #1 Comment #1**, we have updated all the estimates in the revised manuscript and
 removed all the claims about early TB progression has a stronger genetic basis than
 population-wide TB susceptibility. In particular we have added the following text in the main text:

“Using GCTA, we estimated the h_g^2 of population-wide TB susceptibility to be 17.8%
 (s.e.=0.02, $P = 2.85 \times 10^{-21}$) with assumed prevalence of 0.04². Even though the point
 estimate of h_g^2 of TB progression is greater than for population-wide TB risk in the

Russian study, these are not statistically different from each other (two tailed $P = 0.68$,
**Supplementary Figure 6**). Regardless, the strong host genetic basis of TB progression
suggests that larger progression studies may be well-powered to discover additional
variants.”

3. With regard to the case-only analysis it would be interesting to
see the rs73226617 allele frequencies in clustered molecular
fingerprint cases vs. unique molecular fingerprint cases vs. controls.
Might it be informative to present a Bayesian analysis comparing
progressors, reactivators vs shared controls, asking whether the most
likely model is indeed an association restricted to the progressors?

Response:

The MAF of the top associated variant (rs73226617) in clustered molecular fingerprint and
secondary cases is 4.69% and 3.26% in the unique molecular fingerprint (as shown in the
previous response). We have incorporated this information in the updated **Supplementary**
**Figure 9** (see below), where MAF of rs73226617 in the clustered cases is clearly higher than in
the non-clustered cases.

**Supplementary Figure 9. TB cases stratified by a molecular fingerprint.** All cultures of the
cases were genotyped using MIRU-VNTR. TB cases share the same molecular fingerprint are
epidemiologically more related while cases in which fingerprints are unique are due to remote
infection that has reactivated. Reported minor allele frequency (MAF) in each category is of the
top associated variant rs73226617.

Taking the Reviewer's suggestions, we performed a Bayesian analysis to test whether the
reported association is restricted to the early progressors. We calculated the approximate Bayes
factor (ABF)²⁵ for the top associated variant (rs73226617), testing the hypothesis that the
reported association is specific to early progressors with a shared molecular fingerprint. We
assumed the variance σ^2 around the true effect to be 0.04 as suggested by previous
studies^{25,26}. We assumed the probability of correlated true effects (ρ) between two phenotypes
to be 0.5. The disease specific $\log_{10}(ABF)$ (i.e., the ratio of the marginal likelihood for a model
where the variant is only associated with early progressor who has a shared molecular
fingerprints and/or a secondary cases ($\log_{10}(ABF) = 5.81$) and for a model where is associated
with all progressors ($\log_{10}(ABF) = 6.12$) is -0.31. This suggested that the SNP is most likely to
be associated with early progressors who have recent exposure to *M.tb.* alone, but almost
equally likely to be associated with TB progression in general. To test the robustness of the
model using different priors (σ^2 and ρ), we varied the values of $\sigma = \{0.1, 0.2, 0.3, 0.4\}$ and
$\rho = \{0, 0.1, 0.2, 0.3, 0.4, 0.5, 0.6, 0.7, 0.8, 0.9\}$ but did not detect a strong difference that would alter
the conclusion above (new **Supplementary Table 17**).

We are grateful that the Reviewer pointed out the initial mistake in interpreting of our secondary
case-only analysis. We wanted to argue that our reported risk locus is not only associated with
disease progression, but also associated with the early progression after recent exposure to
*M.tb.* This does not mean the reported signals are restricted to these early progressors with
shared molecular fingerprints only.

We have corrected the interpretation in the following text in the manuscript:

"To determine whether the reported risk locus at 3q23 also has an independent
association with TB progression from recent *M.tb* infection, we conducted a case-only
analysis removing age from our case selection criteria. ... We next performed a

Bayesian analysis to test whether the reported association is restricted to the early
progressors after recent exposure to *M.tb.* (**Methods**). The disease specific approximate
Bayes Factor²⁵ (i.e., the ratio of the marginal likelihood for a model where the variant is
only associated with early progressor who has a shared molecular fingerprints and/or a
secondary cases and for a model where is associated with all progressors) is 0.42. This
suggested that the SNP is most likely to be associated with early progressors who have
recent exposure to *M.tb.* alone, but almost equally likely to be associated with TB
progression in general..”

4. The functional data highlights that the locus sits in a regulatory
region in a plausible cell type, but does little to move forward our
understanding of the biology underlying any TB association at 3q23. At
a minimum, eQTL data supporting a cis association would be very
helpful advancing our understanding of any disease association.

**Response:**

As suggested, we took several approaches to identify functional association with the reported
risk locus.

As per **Reviewer #1 Comment #6**, to strengthen our understanding of the biological
implications of the novel risk locus, we first looked for cell-type-specific regulatory elements
using an updated version of IMPACT⁹. Briefly, IMPACT learns an epigenomic signature at
cell-type-specific transcription factor binding sites in a logistic regression framework. Each
variant is assigned with a probability between 0 (least likely to be a regulatory element) and 1
(most likely to be a regulatory element). We computed the probability of a cell-type-specific
regulatory element separately for 15 different cell types and cell states of which 10 are immune
cell types with known roles in TB outcomes with a known role in TB outcomes, including T cells,
B cells, monocytes, macrophages, and peripheral blood cells (new **Figure 3e, per Reviewer #1**
**Comment #6**). To this end, we downloaded publicly available ChIP-seq experiments of
canonical core transcription factors for each cell type separately. We observed
monocyte-specific predicted regulatory elements at rs73226617 at chr3:141400653 and
rs148722713 at chr3:141401146 (IMPACT score 0.79 and 0.41 respectively).

Next, we searched public promoter Hi-C databases¹⁰ to identify any significant interactions
between the 11 risk variants and their potential target genes. We found 5 out of the 11 risk
variants (rs73226617, rs148722713, rs73226619, rs112304167 and rs146526750) have
regulatory interaction with *ATP1B3* in monocytes (new **Supplementary Figure 15**).

Interestingly, similar to the IMPACT results, we found the variant-gene interactions are strongest
in monocytes compared to other cell types, suggesting cell-type-specific activities in the
identified TB risk locus. *ATP1B3* (ATPase Na⁺/K⁺ Transporting Subunit Beta 3) is a
protein-coding gene. The protein encoded by this gene belongs to the family of Na⁺/K⁺ and
H⁺/K⁺ ATPases beta chain proteins, and to the subfamily of Na⁺/K⁺ -ATPases. Na⁺/K⁺
-ATPases are composed of an alpha, beta, and FXYD subunits, and are integral membrane
proteins responsible for establishing and maintaining the electrochemical gradients of sodium
and potassium ions across the plasma membrane through active transport of 3 sodium ions
outside the cell and 2 potassium ions inwards.

It has been reported that *ATP1B3*, along with several genes coding for alpha and beta subunits
are differentially expressed during the course of TB progression after exposure to *Mtb* in a
longitudinal cohort of African household contacts of TB cases¹³. The convergence of the
association with the 3q23 variants with the promoter of *ATP1B3*, as one member of the
Na⁺/K⁺-ATPase family and overall dysregulation of the expression of Na⁺/K⁺-ATPase subunits
during TB progression both suggest that the *ATP1B3* gene is a likely target of the risk locus
identified in our GWAS analysis. However, the exact cell type, and context in which this gene is
activated remain unresolved using the approaches we applied in this manuscript, and will be
pursued in future studies.

We did not identify any significant eQTL using public databases. This might reflect the fact that
our reported TB risk locus has specific activities in monocytes, under specific cell-contexts or
stimulation conditions, or in other non-immune cells, but not in other cell types. However, most
of the public eQTL were reported in non-monocyte cell lines or have limited sample size²⁷⁻²⁹. In
addition, large scale gene expression studies such as the GTEx project¹⁶ reported that
less than 20% of complex trait associated loci have a cis-eQTL overlap. Therefore the lack of
eQTL signals of our reported risk locus is not a surprising result.

To further strengthen our understanding, we searched for other epigenomic evidence that may
indicate changes at transcriptional enhancers and other cis-regulatory elements. Having
previous knowledge of monocyte-specific activity of the identified risk locus, we actively sought
datasets that include monocyte cell-lines. We used data presented in the BLUEPRINT project¹⁴
to search for chromQTLs (**Methods**). We observed significant chromQTL present in the region
(characterized by the presence of H3K4me1) in monocyte (new **Supplementary Figure 13**),
further supporting the idea that this region is indeed an enhancer. All four SNPs that were
included in the dataset are in high LD with the top associated chromQTL signal (rs1568171, $D' =$
1.0) .

Together, this evidence strongly supports that our identified TB risk locus harbors
monocyte-specific predicted regulatory elements. We next performed a CRISPR/Cas9
experiment to test the hypothesis that the 3q23 variants marked an enhancer haplotype where
we expect gene regulatory activities based on epigenetic features. Due to the well-documented
difficulties of CRISPR/Cas9 editing primary human monocytes, we investigated these loci in
THP1 cells, a well-studied monocytic cell line. We disrupted the enhancer region by introducing
insertions/deletions and measured the expression of eight genes in the 0.5 MB surrounding the
top variant by low-input RNA sequencing. Among the eight tested genes, we did not detect any
statistically different expression level before and after disrupting the enhancer region (new
**Supplementary Figure 17**). This might be due to the chosen cell line (THP1) being unable to
completely reflect primary cell biology. In particular we noted the enhancer activity seen in
primary monocytes, is not seen in THP1 cell lines suggested by public ChIP-seq and ATAC-seq
databases^{19–22} (new **Supplementary Figure 18**).

We amended the manuscript to reflect these additional evidence and experiments (**Line**
**243-311**). Together with five new figures including: **Figure 3e** (IMPACT analysis)
**Supplementary Figure 13** (chromQTL), **Supplementary Figure 15** (Promoter capture Hi-C),
**Supplementary Figure 16** (CRISPR/Cas9 experiment), **Supplementary Figure 17** (low-input
RNA-sequencing analysis) and **Supplementary Figure 18** (enhancer activity in primary
monocytes and THP1 cell lines).

We hope these new lines of evidence can increase the Reviewer and readers' confidence in our
finding, and move forward our understanding of the biology underlying the reported association.

Minor Points

1. The sentence “We quantified...” on lines 97–98 seems redundant.

**Response:**

We have **removed** the sentence.

~~“We quantified h_g^2 of TB progression and observed a surprisingly strong genetic basis.”~~

2. The study reports using SNP2HLA in the methods, but these results
don’t appear to be in the results. It would be of interest to report
the associated classical HLA allele/amino acids at the class I locus.

**Response:**

We thank the Reviewer for this suggestion. We expanded our results section to report more
details of the classical HLA allele/amino acid association in the main text:

“Next, we performed an HLA imputation using a multi-ethnic HLA reference panel
(**Methods**), and obtained genotypes for classical alleles as well as amino acid positions
of three class I (*HLA-A*, *HLA-B*, *HLA-C*) and three class II (*HLA-DQA1*, *HLA-DQB1*,
*HLA-DRB1*) HLA genes. Using the same linear mixed model framework (**Methods**,
**Supplementary Figure 12**), we tested associations between specific amino acid
positions and TB progression which identified the most significant association at amino
acid position 73 of *HLA-A* (OR=1.12, $P = 1.03 \times 10^{-6}$). We noted several other amino
acids of class I genes with suggestive associations ($P < 1 \times 10^{-5}$), including position 97
of *HLA-B* (OR=1.05, $P = 8.99 \times 10^{-6}$). Notably, amino acid variability at this position
affects the structure and flexibility of the peptide binding groove and is associated with
many infectious and autoimmune phenotypes, such as HIV-1 viral load^{30,31} and
ankylosing spondylitis³². These results suggest that HLA class I genes might play a role
in TB progression.”

We updated **Supplementary Figure 12** for this analysis:

**Supplementary Figure 12. Manhattan plot of HLA region.** We imputed HLA region using
SNP2HLA with a multi-ethnic HLA reference panel. The most significant amino acid association
is position 73 of HLA-A (OR=1.12, $P = 1.03 \times 10^{-6}$)

3. Line 284-285: "Sex and age were included as fixed effects to
correct for population stratification (Supplementary Figure 2)." The
supplementary figure this refers to is presumably S3? Also I assume
that this sentence is missing inclusion of principal components 1 and
2 (or was the GRM alone use to control for population structure)?
Either seems appropriate, this just wasn't clear to me.

**Response:**

We thank the Reviewer for pointing out this issue. We have now changed the text to:

"We used the genetic relatedness matrix (GRM) as random effects to correct for cryptic
relatedness and population stratification between collected individuals. Sex and age
were included as fixed effects."

3. Line 241-2 methods - it reads as if cases in the main GWAS had to
have an M.tb isolate shared with another case: "(2) index patients

whose M. tb isolates shared a molecular fingerprint with isolates from
other enrolled patients". But then why are there cases with unique M.
tb molecular fingerprints in the case-only analysis?

**Response:**

We thank the Reviewer for pointing out this possible confusion. We clarify that our three
selection criteria were not dependent on each other. That is, if an individual satisfies one of the
three three conditions, it will be enrolled in the GWAS as a case. We reflect this in the main
manuscript with the following text:

"All cases were HIV-negative, culture-positive and drug-sensitive who have pulmonary
TB. We defined cases who were likely to have recently exposed TB, if a case satisfied at
least one of the three criteria: (1) exposed HHCs who developed active TB during a 12
848 month follow up period; (2) index patients whose M.tb isolates shared a molecular
fingerprint with isolates from other enrolled patients and (3) index patients who were 40
850 years old or younger at time of diagnosis."

4. The (negative) rare variant result should be reported in the body
of the main text.

**Response:**

We thank the Reviewer for the suggestion, and reported the rare variant result in the body of the
main text:

"We observed no significant rare variant (minor allele frequency (MAF) <1%) association
with TB progression after performing gene-based generalized linear mixed model
(Methods)."

- 1. Curtis, J. *et al.* Susceptibility to tuberculosis is associated with variants in the ASAP1 gene
encoding a regulator of dendritic cell migration. *Nat. Genet.* **47**, 523–527 (2015).
- 2. Speed, D. *et al.* Reevaluation of SNP heritability in complex human traits. *Nat. Genet.* **49**,
986–992 (2017).

- 3. Kanai, M., Tanaka, T. & Okada, Y. Empirical estimation of genome-wide significance
thresholds based on the 1000 Genomes Project data set. *J. Hum. Genet.* **61**, 861–866
(2016).
- 4. Stead, W. W. & To, T. The significance of the tuberculin skin test in elderly persons. *Ann.*
*Intern. Med.* **107**, 837–842 (1987).
- 5. Lux, M. Perfect subjects: race, tuberculosis, and the Qu'Appelle BCG Vaccine Trial. *Can.*
*Bull. Med. Hist.* **15**, 277–295 (1998).
- 6. Greenwood, C. M. *et al.* Linkage of tuberculosis to chromosome 2q35 loci, including
NRAMP1, in a large aboriginal Canadian family. *Am. J. Hum. Genet.* **67**, 405–416 (2000).
- 7. Jones, D. S. Virgin Soils Revisited. *William Mary Q.* **60**, 703–742 (2003).
- 8. McMillen, C. W. 'The Red Man and the White Plague': Rethinking Race, Tuberculosis, and
American Indians, ca. 1890–1950. *Bull. Hist. Med.* **82**, 608–645 (2008).
- 9. Amariuta, T. *et al.* IMPACT: Genomic Annotation of Cell-State-Specific Regulatory Elements
Inferred from the Epigenome of Bound Transcription Factors. *Am. J. Hum. Genet.* (2019).
doi:10.1016/j.ajhg.2019.03.012
- 10. Schofield, E. C. *et al.* CHiCP: a web-based tool for the integrative and interactive
visualization of promoter capture Hi-C datasets. *Bioinformatics* **32**, 2511–2513 (2016).
- 11. Javierre, B. M. *et al.* Lineage-Specific Genome Architecture Links Enhancers and
Non-coding Disease Variants to Target Gene Promoters. *Cell* **167**, 1369–1384.e19 (2016).
- 12. Takheaw, N. *et al.* Ligation of Na, K ATPase β 3 subunit on monocytes by a specific
monoclonal antibody mediates T cell hypofunction. *PLoS One* **13**, e0199717 (2018).
- 13. Duffy, F. J. *et al.* Immunometabolic Signatures Predict Risk of Progression to Active
Tuberculosis and Disease Outcome. *Front. Immunol.* **10**, 527 (2019).
- 14. Chen, L. *et al.* Genetic Drivers of Epigenetic and Transcriptional Variation in Human

- Immune Cells. *Cell* **167**, 1398–1414.e24 (2016).
- 15. Dimas, A. S. *et al.* Common Regulatory Variation Impacts Gene Expression in a Cell
Type-Dependent Manner. *Science* **325**, 1246–1250 (2009).
- 16. The GTEx Consortium. The Genotype-Tissue Expression (GTEx) pilot analysis: Multitissue
gene regulation in humans. *Science* **348**, 648–660 (2015).
- 17. Gutierrez-Arcelus, M. *et al.* Tissue-specific effects of genetic and epigenetic variation on
gene regulation and splicing. *PLoS Genet.* **11**, e1004958 (2015).
- 18. Gutierrez-Arcelus, M., Baglaenko, Y., Arora, J. & Hannes, S. Allele-specific expression
changes dynamically during T cell activation in HLA and other autoimmune loci. *bioRxiv*
(2019).
- 19. Corces, M. R. *et al.* Lineage-specific and single-cell chromatin accessibility charts human
hematopoiesis and leukemia evolution. *Nat. Genet.* **48**, 1193–1203 (2016).
- 20. Phanstiel, D. H. *et al.* Static and Dynamic DNA Loops form AP-1-Bound Activation Hubs
during Macrophage Development. *Mol. Cell* **67**, 1037–1048.e6 (2017).
- 21. Mohaghegh, N. *et al.* NextPBM: a platform to study cell-specific transcription factor binding
and cooperativity. *Nucleic Acids Res.* **47**, e31 (2019).
- 22. ENCODE Project Consortium. An integrated encyclopedia of DNA elements in the human
genome. *Nature* **489**, 57–74 (2012).
- 23. Adhikari, K. *et al.* A GWAS in Latin Americans highlights the convergent evolution of lighter
skin pigmentation in Eurasia. *Nat. Commun.* **10**, 358 (2019).
- 24. Yablonskii, P. K., Vigel, A. A., Galkin, V. B. & Shulgina, M. V. Tuberculosis in Russia. Its
history and its status today. *Am. J. Respir. Crit. Care Med.* **191**, 372–376 (2015).
- 25. Wakefield, J. Bayes factors for genome-wide association studies: comparison with
P-values. *Genet. Epidemiol.* **33**, 79–86 (2009).

- 26. Jostins, L. & McVean, G. Trinculo: Bayesian and frequentist multinomial logistic regression
for genome-wide association studies of multi-category phenotypes. *Bioinformatics* **32**,
1898–1900 (2016).
- 27. Fairfax, B. P. *et al.* Innate immune activity conditions the effect of regulatory variants upon
monocyte gene expression. *Science* **343**, 1246949 (2014).
- 28. Raj, T. *et al.* Polarization of the effects of autoimmune and neurodegenerative risk alleles in
leukocytes. *Science* **344**, 519–523 (2014).
- 29. Nagai, A. *et al.* Overview of the BioBank Japan Project: Study design and profile. *J.*
*Epidemiol.* **27**, S2–S8 (2017).
- 30. International HIV Controllers Study *et al.* The major genetic determinants of HIV-1 control
affect HLA class I peptide presentation. *Science* **330**, 1551–1557 (2010).
- 31. McLaren, P. J. *et al.* Polymorphisms of large effect explain the majority of the host genetic
contribution to variation of HIV-1 virus load. *Proc. Natl. Acad. Sci. U. S. A.* **112**,
14658–14663 (2015).
- 32. Cortes, A. *et al.* Major histocompatibility complex associations of ankylosing spondylitis are
complex and involve further epistasis with ERAP1. *Nat. Commun.* **6**, 7146 (2015).
- 30. Cortes, A. *et al.* Major histocompatibility complex associations of ankylosing spondylitis are
complex and involve further epistasis with ERAP1. *Nat. Commun.* **6**, 7146 (2015).

Reviewers' Comments:

Reviewer #1:

Remarks to the Author:

I appreciate the careful and thorough responses of the authors. I feel they have substantially addressed the previous concerns through multiple additional analyses and experiments that improve the quality of the manuscript. This is an important and valuable study, which may inform future ID GWAS design. While some aspects still aren't entirely convincing, I feel some uncertainty in these kinds of studies with heterogenous populations, exposures, pathogens, etc... are to be expected. The QQ plots, especially when Native American admixture is now considered, do not show much deviation from neutral expectation, but the added permutation analysis at least provides some additional statistical justification for their association based on the calculated significance threshold, and along with the association with cases-only TB progression association ($p=0.02$), there is a fairly good confidence in the association.

I have only a few additional comments as listed below.

1) For the CRISPR/Cas9 experiments, the authors state none of the genes showed an association, but Figure S17 shows an association with GRK7. This would conflict with the ATP1B3 hypothesis and needs to be commented on in the manuscript.

2) Related, I think the IMPACT analysis, promoter capture Hi-C, EMSA in THP-1, and CRISPR/Cas9 experiments were all worthwhile to carry out, but the assignment of causal SNP and especially gene are still not fully convincing. I am not requesting additional experiments, as I feel they have done what can be reasonably expected. I think the conclusion that there are likely monocyte-specific regulatory elements in the region is valid, but the conclusion that ATP1B3 is the causal gene certainly is not well supported. I think it is important to scale-back claims, such as in the Abstract to something like "...ATP1B3 as a plausible target gene..."

3) Thank you for indicating in the response letter that EMSA is based on 3 experiments—I think readers would appreciate having that information in the figure legend for Supp. Figure 14 as well.

4) Complete GWAS summary statistics should be included as a supplemental data file or deposited into a repository such as LDHub or EBI-GWAS Catalog.

Reviewer #3:

Remarks to the Author:

The authors have adequately addressed my comments. The manuscript will be a valuable addition to the field and should be accepted. I have no further substantive concerns.

Minor points.

1. The p-value for the Crohn's h2g on line 153 main text is truncated.

2. In the CRISPR/Cas9 experiment results, the authors report no differential gene expression ($P>0.05$) in any of the 8 genes in cis to rs73226617 (line 307). The data presented in Supplementary Figure 17 suggest that GRK7 expression is significantly increased in edited THP1 clones: data which are excluded from the analysis due to the low expression of GRK7. This is made clear in the methods section, but it would be helpful to also clarify this in the figure legend (or alternatively exclude GRK7 and ZBTB38 from the figure altogether).

3. The authors hypothesise on lines 215-6 that the lack of replication in the Russian and Icelandic GWAS may be explained by: "The association signals were therefore most likely diluted due to the inclusion of reactivation cases and noninfected controls in the cohort collection." While inclusion of

non-infected controls in the Russian data may, in part, underlie dilution of a progression-specific signal in that dataset, the Icelandic GWAS controls included here were TST+. It would benefit the manuscript to have a more complete discussion of the potential reasons for lack of replication in Iceland.

REVIEWERS' COMMENTS:

Reviewer #1 (Remarks to the Author):

I appreciate the careful and thorough responses of the authors. I feel they have substantially addressed the previous concerns through multiple additional analyses and experiments that improve the quality of the manuscript. This is an important and valuable study, which may inform future ID GWAS design. While some aspects still aren't entirely convincing, I feel some uncertainty in these kinds of studies with heterogenous populations, exposures, pathogens, etc... are to be expected. The QQ plots, especially when Native American admixture is now considered, do not show much deviation from neutral expectation, but the added permutation analysis at least provides some additional statistical justification for their association based on the calculated significance threshold, and along with the association with cases-only TB progression association ($p=0.02$), there is a fairly good confidence in the association.

I have only a few additional comments as listed below.

1) For the CRISPR/Cas9 experiments, the authors state none of the genes showed an association, but Figure S17 shows an association with GRK7. This would conflict with the ATP1B3 hypothesis and needs to be commented on in the manuscript.

Response: The expression level of GRK7 was too low ($TPM < 1$) to confidently ascribe any biological significance to the differential expression. The following sentence was already included in the Method section entitled *Gene expression analysis of edited THP1 cells*: "We considered as expressed genes those with a $\log_2(TPM+1) > 1$ in at least 95% of the samples". To avoid confusion, we have excluded *GRK7* and *ZBTB38* from **Supplementary Figure 17a** and added the following text in the legend to clarify:

"Expression of six genes around rs73226617 with transcripts per million (TPM) >1 in THP1 clones, which maintained wildtype genomic sequence after expansion of single cells from bulk-edited THP1 cells compared to edited clones. P-values are derived from a linear regression model including first principal component of the gene expression profile as covariate. "

2) Related, I think the IMPACT analysis, promoter capture Hi-C, EMSA in THP-1, and CRISPR/Cas9 experiments were all worthwhile to carry out, but the assignment of causal SNP

and especially gene are still not fully convincing. I am not requesting additional experiments, as I feel they have done what can be reasonably expected. I think the conclusion that there are likely monocyte-specific regulatory elements in the region is valid, but the conclusion that ATP1B3 is the causal gene certainly is not well supported. **I think it is important to scale-back claims, such as in the Abstract to something like “...ATP1B3 as a plausible target gene...”**

Response: We have modified the abstract text per the Reviewer’s suggestion:

“With *in silico* and *in vitro* analyses we identify rs73226617 or rs148722713 as the likely functional variant and *ATP1B3* as a potential causal target gene with monocyte specific function.”

3) Thank you for indicating in the response letter that EMSA is based on 3 experiments—I think readers would appreciate having that information in the figure legend for Supp. Figure 14 as well.

Response: We added the following sentence to the legend of **Supplementary Figure 14**:
“The experiment was performed on three independent batches of THP1 nuclear extracts”

4) Complete GWAS summary statistics should be included as a supplemental data file or deposited into a repository such as LDHub or EBI-GWAS Catalog.

Response: We have deposited the data. Summary statistics will be made available through the NHGRI-EBI GWAS Catalog <https://www.ebi.ac.uk/gwas/downloads/summary-statistics>. And this information is provided in the “Data Availability” section.

Reviewer #3 (Remarks to the Author):

The authors have adequately addressed my comments. The manuscript will be a valuable addition to the field and should be accepted. I have no further substantive concerns.

Minor points.

1. The p-value for the Crohn’s h2g on line 153 main text is truncated.

Response: We thank the Reviewer for noticing this, and has addressed this issue.

2. In the CRISPR/Cas9 experiment results, the authors report no differential gene expression ($P > 0.05$) in any of the 8 genes in cis to rs73226617 (line 307). The data presented in Supplementary Figure 17 suggest that GRK7 expression is significantly increased in edited THP1 clones: data which are excluded from the analysis due to the low expression of GRK7. This is made clear in the methods section, but it would be helpful to also clarify this in the figure legend (or alternatively exclude GRK7 and ZBTB38 from the figure altogether).

Response: We have taken the Reviewer's suggestion and excluded *GRK7* and *ZBTB38* from the figure altogether (see below).

Supplementary Figure 17. Low-input RNA-seq analysis.

(a) Expression of six genes around rs73226617 with transcripts per million (TPM) > 1 in THP1 clones, which maintained wildtype genomic sequence after expansion of single cells from bulk-edited THP1 cells compared to edited clones. P-values are derived from a linear regression model including first principal component of the gene expression profile as covariate. (b) Volcano plot from RNA-seq data showcasing global expression of transcripts enriched in wildtype (left, $n=7$) or edited (right, $n=16$) THP1 clones. Source data are provided as a Source Data file.

3. The authors hypothesise on lines 215-6 that the lack of replication in the Russian and Icelandic GWAS may be explained by: “The association signals were therefore most likely diluted due to the inclusion of reactivation cases and noninfected controls in the cohort collection.” While inclusion of non-infected controls in the Russian data may, in part, underlie dilution of a progression-specific signal in that dataset, the Icelandic GWAS controls included here were TST+. It would benefit the manuscript to have a more complete discussion of the potential reasons for lack of replication in Iceland.

Response: We suspect the lack of association in Iceland is partially due to the inclusion of reactivation cases. However, other reasons are also plausible, such as difference in TB prevalence between different populations. We thus added the following sentence to the main text to have a more complete discussion of the potential reasons for lack of replication in both cohort.

“The lack of association observed in the European cohort could be due to the inclusion of reactivation TB cases and noninfected controls in the cohort collection and/or differences in TB prevalence (**Methods**).”